# Future surface mass balance and surface melt in the Amundsen sector of the West Antarctic Ice Sheet

Marion Donat-Magnin[1], Nicolas C. Jourdain[1], Christoph Kittel[2], Cécile Agosta[3], Charles Amory[2], Hubert Gallée[1], Gerhard Krinner[1], and Mondher Chekki[1]

[1]Univ. Grenoble Alpes/CNRS/IRD/G-INP, IGE, Grenoble, France
[2]F.R.S.-FNRS, Laboratory of Climatology, Department of Geography, University of Liège, B-4000 Liège, Belgium
[3]Laboratoire des Sciences du Climat et de l'Environnement, LSCE-IPSL, CEA-CNRS-UVSQ Université Paris-Saclay, F-91198 Gif-sur-Yvette, France

**Correspondence:** Nicolas C. Jourdain (nicolas.jourdain@univ-grenoble-alpes.fr)

**Abstract.** We present projections of West-Antarctic surface mass balance (SMB) and surface melt to 2080-2100, under the RCP8.5 scenario and based on a regional model at 10 km resolution. Our projections are built by adding a CMIP5 (5[th] Coupled Model Intercomparison Project) multi-model-mean seasonal climate-change anomaly to the present-day model boundary conditions. Using an anomaly has the advantage to reduce CMIP5 model biases, and a perfect-model test reveals that our approach captures most characteristics of future changes, despite a 16-17% underestimation of projected SMB and melt rates.

SMB over the grounded ice sheet in the sector between Getz and Abbot increases from $336\,\mathrm{Gt\,yr^{-1}}$ in 1989-2009 to $455\,\mathrm{Gt\,yr^{-1}}$ in 2080-2100, which would reduce the global sea level changing rate by $0.33\,\mathrm{mm\,yr^{-1}}$. Snowfall indeed increases by 7.4 to 8.9% per °C of near-surface warming, due to increasing saturation water vapour pressure in warmer conditions, reduced sea-ice concentrations, and more marine air intrusion.

Ice-shelf surface melt rates increase by an order of magnitude in the 21[st] century, mostly due to higher downward radiation from increased humidity, and to reduced albedo in the presence of melting. There is a net production of surface liquid water over eastern ice shelves (Abbot, Cosgrove and Pine Island) but not over western ice shelves (Thwaites, Crosson, Dotson and Getz). This is explained by the evolution of the melt-to-snowfall ratio: below a threshold of 0.60 to 0.85 in our simulations, firn air is not entirely depleted by melt water, while entire depletion and net production of surface liquid water occur for higher ratios. This suggests that western ice shelves might remain unaffected by hydrofracturing for more than a century under RCP8.5, while eastern ice shelves have a high potential for hydrofracturing before the end of this century.

## 1 Introduction

In a perfectly stable climate, the Antarctic ice sheet would have a constant mass, and the Surface Mass Balance (SMB, the sum of rainfall and snowfall minus sublimation, runoff and eroded snow) over the grounded ice sheet, i.e. 2000 to $2100\,\mathrm{Gt\,yr^{-1}}$

under the present climate (van Wessem et al., 2018; Agosta et al., 2019; Mottram et al., 2020), would be exactly compensated by the ice flow across the grounding line, i.e., into the ocean. In contrast to this hypothetical stable climate, the Antarctic ice sheet has lost $2720\pm1390$ Gt of grounded ice from 1992 to 2017, which corresponds to $7.6\pm3.9$ mm of sea level rise (Shepherd et al., 2018). The main origin of the current mass loss is the acceleration of major ice streams (Bamber et al., 2018; Shepherd et al., 2018; Rignot et al., 2019), mostly driven by increased oceanic melt (e.g. Turner et al., 2017; Jenkins et al., 2018). In a warmer climate, SMB may significantly increase and partly compensate the loss due to accelerated ice flows (e.g. Favier et al., 2017; Seroussi et al., 2020).

Recent SMB trends (1979-2000), reconstructed from firn cores, are slightly negative, with SMB decreasing at a rate of $-2.7\pm3.8$ Gt yr$^{-2}$ for the entire ice sheet, and large differences between West Antarctica ($-0.1\pm1.4$ Gt yr$^{-2}$) and East Antarctica ($-4.5\pm3.5$ Gt yr$^{-2}$) (Medley and Thomas, 2019). These trends only represent a small fraction of the current ice-sheet imbalance ($-110$ Gt yr$^{-1}$ over 1992-2017), but climate models predict a significant increase in Antarctic precipitation over the 21st century (Krinner et al., 2007; Genthon et al., 2009; Nowicki et al., 2020). Under the A1B or RCP8.5 scenarios to 2100, simulated Antarctic SMB increases by 13 to 25% depending on the climate model (Agosta et al., 2013; Ligtenberg et al., 2013; Lenaerts et al., 2016; Palerme et al., 2017). These changes are strongly related to temperatures (Frieler et al., 2015), with an SMB sensitivity of 5-7% per degree of near-surface warming (Krinner et al., 2008; Ligtenberg et al., 2013; Agosta et al., 2013; Palerme et al., 2017). In a warmer climate, the saturation water vapour pressure indeed increases (Clapeyron, 1834; Clausius, 1850), and more humidity can be transported over the ocean and made available for Antarctic precipitation.

Runoff into the ocean is a negative contribution to SMB. It is produced if surface melt and/or rain rates are high enough to (i) bring the temperature of underlying snow/firn layers to the freezing point, and (ii) percolate and saturate the pore space in the snow and firn layers, which is sometimes referred to as firn air depletion (Pfeffer et al., 1991; Kuipers Munneke et al., 2014; Alley et al., 2018). Currently, runoff is at least three orders of magnitude smaller than precipitation for the entire Antarctic ice sheet (van Wessem et al., 2018; Agosta et al., 2019). Surface melt and/or rainfall beyond the pore space saturation do not necessarily lead to runoff. Liquid water in excess can alternatively be transported horizontally on the ice shelf and/or form ponds. In some circumstances, these processes can trigger ice-shelf collapse through hydrofracturing or ice-shelf bending (e.g. Bell et al., 2018; Robel and Banwell, 2019; Lai et al., 2020), with important consequences for the ice sheet dynamics (Sun et al., 2020).

Surface melt only occurs to a significant extent over the Peninsula (Scambos et al., 2009; Trusel et al., 2013), and sporadically in other regions like the Amundsen Sea (Nicolas et al., 2017; Donat-Magnin et al., 2020) and Ross Sea (Bell et al., 2017; Wille et al., 2019) sectors. Over most ice shelves in Antarctica, current melt and rainfall rates are nonetheless low compared to snowfall rates, and there is no significant firn air depletion (Kuipers Munneke et al., 2014). In such circumstances, the resulting liquid water is buried in the snow and likely refreezes. An exception is the Eastern side of the Antarctic Peninsula that experienced melt water ponds and the resulting collapse of Larsen B ice shelf (Vaughan et al., 2003; van den Broeke, 2005; Scambos et al., 2009), and where simulations suggest nearly depleted firn air (Kuipers Munneke et al., 2014).

In a warmer climate, surface melt increases exponentially with surface air temperature (Kuipers Munneke et al., 2014; Trusel et al., 2015), potentially leading to higher mass loss through runoff and to the occurrence of melt water ponds, ice-

shelf hydrofracturing or bending and subsequent ice-shelf collapse. Based on a firn model forced by regional atmospheric simulations constrained by global projections, Kuipers Munneke et al. (2014) estimated that a few more ice shelves could experience near-complete firn air depletion by 2100, and many more ice shelves by 2200 in East and West Antarctica under the strongest emission scenario. Using regional atmospheric simulations and global projections with bias corrections, Trusel et al. (2015) reported that large fractions of East and West Antarctic ice shelves could experience melt rates greater than the pre-collapse value of Larsen B by the end of the 21st century under the warmest scenario.

Computing projections of future SMB and surface melt rates remains challenging, because of the strong natural variability at regional scales (Lenaerts et al., 2016; Donat-Magnin et al., 2020), biases in global climate models (GCMs) (Bracegirdle et al., 2013; Swart and Fyfe, 2012) and GCM resolutions that are too coarse to resolve the orographic processes in the relatively steep coastal area (Krinner et al., 2008; Lenaerts et al., 2012; Agosta et al., 2013). Most models that participated in the 5th Climate Model Intercomparison Project (CMIP5 Taylor et al., 2012) overestimated the present-day Antarctic precipitation, by more than 100% in some cases (Palerme et al., 2017). These models also had a generally poor representation of the snow-pack energy balance, which is why future melt rate estimates have often been derived from simulated air temperatures rather than directly provided by the models (Davies et al., 2014; Trusel et al., 2015), and most limitations remain relevant in the CMIP6 models (Mudryk et al., 2020). Recent versions of regional climate models (RCMs) with a comprehensive representation of polar processes are now able to simulate melt rates in reasonable agreement with observational estimates when they are driven by reanalyses (van Wessem et al., 2018; Lenaerts et al., 2018; Datta et al., 2019; Donat-Magnin et al., 2020). Using this kind of RCMs to downscale simulations from GCMs can significantly reduce surface biases (Fettweis et al., 2013). However, this approach is not sufficient to remove the large-scale biases inherited from GCMs, and bias corrections of the GCM solution (Trusel et al., 2015; Beaumet et al., 2019a) may be needed prior to RCM downscaling. In this paper, we build SMB and surface melt projections at the end of the 21st century by forcing an RCM with the 3-dimensional climate-change anomalies from a CMIP5 RCP8.5 multi-model mean, with the aim of removing a part of the CMIP model biases (see section 2).

We focus on the Amundsen Sea sector, where potential future melt-induced hydrofracturing and associated loss of ice-shelf buttressing could have large effects on the stability of the West Antarctic ice sheet and therefore on sea level rise (Pattyn et al., 2019). Currently the Amundsen sector accounts for 60% of the total Antarctic mass loss (Rignot et al., 2019). While oceanic melting is currently the dominant process causing mass loss (Thoma et al., 2008; Turner et al., 2017; Jenkins, 2016; Jenkins et al., 2018), surface air temperature is expected to increase (Bracegirdle et al., 2008), and whether the ice shelves of the Amundsen sectors will respond with the same hydrofracturing mechanism as in the Antarctic Peninsula remains an open question. Contrasting behaviours were indeed projected for individual ice shelves in previous studies at relatively coarse resolution (Kuipers Munneke et al., 2014; Trusel et al., 2015). In the following, we describe our general methodology (section 2), then we describe future projections in section 3, with a particular focus on SMB over the grounded ice sheet (relevant for sea level) and melting over the ice shelves (relevant for hydrofracturing). We then propose an extrapolation of our results to other scenarios and time horizons and we thoroughly discuss the impact of modelling and methodological biases in our projections (section 4).

## 2 Method

### 2.1 Regional atmosphere and firn model

Our projections of the West Antarctic surface climate for the end of the 21[st] century are based on version 3.9.3 of the MAR regional atmospheric model (Gallée and Schayes, 1994; Agosta et al., 2019). Our regional configuration is centred on the Amundsen Sea sector, covers 2800×2400 km, and was developed by Donat-Magnin et al. (2020). The horizontal resolution is 10 km and and we use 24 vertical sigma levels located from approximately 1 m above the ground to 0.1 hPa. The topography is derived from BEDMAP2 (Fretwell et al., 2013) and the drainage basins used for averages are those defined by Mouginot et al. (2017).

The radiative scheme and cloud properties are the same as in Datta et al. (2019) and the surface scheme, including snow density and roughness, are the same as in Agosta et al. (2019). The atmosphere is coupled to the SISVAT surface scheme (Soil Ice Snow Vegetation Atmosphere Transfer, Gallée and Duynkerke, 1997; Gallée et al., 2001), which here is a 30-layer snow/firn model representing the first 20 m with refined resolution at the surface. It includes prognostic equations for temperature, mass, water content and snow properties (dendricity, sphericity and grain size). SISVAT and the atmosphere are coupled through exchanges of mass fluxes as well as radiative and turbulent heat fluxes.

Surface albedo depends on the evolving snow properties and on the solar zenithal angle (Tedesco et al., 2016). As in Agosta et al. (2019), the density of fresh snow increases with wind speed ($+6.84 \, \mathrm{kg \, m^{-3} \, (m \, s^{-1})^{-1}}$) and temperature ($+0.48 \, \mathrm{kg \, m^{-3} \, {}^{\circ}C^{-1}}$) at the time of snow deposit, and our set-up does not include drifting snow, which is not considered as a strong limitation for this sector of Antarctica (Lenaerts et al., 2012) although this remains a poorly quantified process. In case of surface melt or rainfall, liquid water percolates downward into the next firn layers, with a water retention of 10% of the porosity in each successive layer. The firn layers are fully permeable until they reach a close-off density of $830 \, \mathrm{kg \, m^{-3}}$. To account for possible cracks in ice lenses and moulins, the part of available water that is transmitted downward to the next layer decreases as a linear function of firn density, from 100% transmitted at the close-off density to zero at $900 \, \mathrm{kg \, m^{-3}}$ and denser. If the liquid water is not able to percolate further down, then it fills the entire porosity space of surface layers, and the excess is removed from the simulated snow/firn because there is no representation of ponds or horizontal routing of liquid water in our version of MAR. In this paper, we refer to the formation rate of liquid water in excess as "net production of surface liquid water" and we use it as an indicator of potential ice shelf collapse. Liquid water in excess can indeed accumulate in ponds or flow into crevasses (potentially inducing hydrofracturing) or into the ocean (potentially inducing ice-shelf bending). It is important to keep in mind that this is only a potential for collapse because the liquid water can flow into the ocean in some cases (thereby protecting the ice shelf from hydrofracturing; Bell et al., 2017) and because there are mechanical conditions for hydrofracturing to develop (Lai et al., 2020).

### 2.2 Present-day and future forcing

The simulation representative of the present climate is the one described in Donat-Magnin et al. (2020). It is forced laterally (pressure, wind, temperature, specific humidity), at the top four levels (temperature, wind), and at the surface (sea ice con-

centration, sea surface temperature) by 6-hourly outputs of the ERA-interim reanalysis (Dee et al., 2011), which has a good representation of the Antarctic climate (Bromwich et al., 2011; Huai et al., 2019). A thorough evaluation of the present-day simulation with respect to in-situ and satellite observational products is provided in Donat-Magnin et al. (2020) and indicates a good fidelity of the simulated surface mass balance and melt rates. Here we further show that simulated present SMB values over the grounded parts of Pine Island and Thwaites (Tab. 1) are within the observational range of uncertainty estimated by Medley et al. (2014) after correction to account for different basin areas (their Tab. 3). In the present paper, we do not describe or discuss features located eastward of 75°W (e.g. Georges VI ice shelf) and southward of 78°S (e.g. Ross and Ronne ice shelves) as these locations are considered too close to the relaxation zone of the model domain where lateral boundary conditions are prescribed.

For the future, we calculate the climate-change absolute anomaly from a CMIP5 multi-model mean (MMM), and we add it to the 6-hourly ERA-interim variables used to drive MAR, i.e. sea surface temperature (SST), sea ice cover (SIC), and 3-dimensional wind velocity, air temperature and specific humidity. Considering all these anomalies together allows keeping the consistency of linear relationships, such as the geostrophic and thermal wind balances, although it does not necessarily conserve non-linear relationships. This type of method was previously referred to as "anomaly nesting" (Misra and Kanamitsu, 2004), or "pseudo global warming" (e.g., Kimura and Kitoh, 2007) although this term was often used for more simple temperature and humidity perturbation methods (e.g. Schär et al., 1996). The MMM anomaly is defined as the mean difference between 1989-2009 and 2080-2100 under the RCP8.5 emission scenario, for an ensemble of 33 CMIP5 models (see Appendix A). The anomaly is calculated separately for each calendar month, meaning that we apply an anomaly that includes a seasonal cycle. Monthly anomalies are linearly interpolated to avoid discontinuity of 6-hourly boundary conditions. In the future simulation, we do not modify greenhouse gases concentrations in our regional domain, which is expected to have a minor effect because the dominant effect of global increase in greenhouse gases concentrations on our regional simulations comes from changes in sea surface and sea ice forcing as well as through increased humidity and temperature at the lateral boundaries (Krinner et al., 2014; Bull et al., 2020).

Adding an anomaly is relatively simple, but requires a specific calculation for two variables. First, specific humidity is set to zero in the rare cases where applying the CMIP5 anomaly would produce unphysical negative values. Second, sea-ice concentration (SIC) anomalies are applied through an iterative process, which is needed because some locations have non-zero SIC on some days, and zero SIC on other days. As negative SIC values are unphysical, applying a negative climatological SIC anomaly to all days (but keeping days with zero SIC unchanged) does not conserve the applied CMIP5 anomaly. To circumvent this issue, we apply the anomaly through 20 iterations: we start applying the CMIP5-MMM anomaly to the days and locations with SIC greater than zero (for negative anomaly) and smaller than 100% (for positive anomaly), and after each iteration, we calculate the residual SIC that would be needed to reach the original CMIP5-MMM SIC anomaly, and we add it to the applied climatological anomaly. The effect of this sea-ice anomaly correction is briefly described in section 4. Alternative sea-ice correction methods were evaluated by Beaumet et al. (2019b), but here we prefer to stay as close as possible to the simple anomaly method used for the other variables.

As discussed by Knutti et al. (2010), the MMM is often considered as the "best" estimate for future climate because individual model biases are partly cancelled in the MMM, although an equal weight for all the models does not account for the fact that models are not independent from each other because of the same operating centres, common history, shared physical parameterisations and numerical methods (Knutti et al., 2017; Herger et al., 2018). Given that the CMIP model biases are largely stationary even under strong climate changes (Krinner and Flanner, 2018), our method is also expected to remove a part of the biases in individual model projections. This method has previously been used in various regional studies (e.g. Sato et al., 2007; Knutson et al., 2008; Michaelis et al., 2017; Dutheil et al., 2019) but, to our knowledge, never in Antarctica. Krinner et al. (2008, 2014) and Beaumet et al. (2019a) used anomalies in global simulations with a stretched grid over Antarctica, but this only involved anomalies in sea surface conditions.

All the simulation years are run in parallel with a 12-year spin up for each simulated year, which is sufficient to obtain a steady net production of surface liquid water in the future simulation over all ice shelves except Abbot (see Discussion). When not stated otherwise, the present-day period represents 1988-2017. The future period corresponds to the 1988-2017 period to which was added the CMIP5-MMM anomaly (2080-2100 minus 1989-2009) and therefore represents something like 2079-2108 (with the interannual variability of 1988-2017). While our CMIP5-MMM anomaly is only based on 21 years, we decided to run our regional simulations over 30 years, which provides more statistical significance given that surface melt rates and SMB exhibit high interannual variability in this region (Scott et al., 2019; Donat-Magnin et al., 2020). Retrospectively, it would have been better to use the same 30-year time window for CMIP5 and for our MAR simulations.

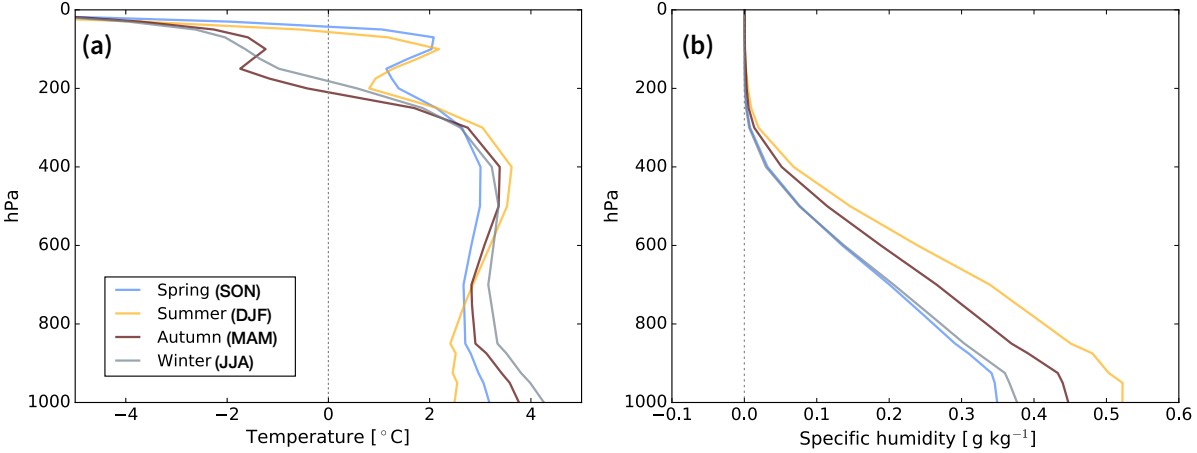

**Figure 1.** (a) temperature and (b) specific humidity vertical profiles of the CMIP5-MMM anomaly (2080-2100 minus 1989-2009) that is added to ERA-interim, here spatially-averaged over West Antarctica (60-85°S , 170-40°W ).

We now briefly describe the CMIP5-MMM anomalies applied to ERA-interim. The troposphere is warmed relatively uniformly from the surface to ∼300 hPa (Fig. 1a). There is a clear seasonal cycle in the low-troposphere anomalies, with stronger warming in winter than in summer. This is related to stronger changes in winter sea-ice cover compared to summer (solid lines in Fig. 2), because present-day summers are already relatively sea-ice free and, as such, sea-ice cover cannot decrease much

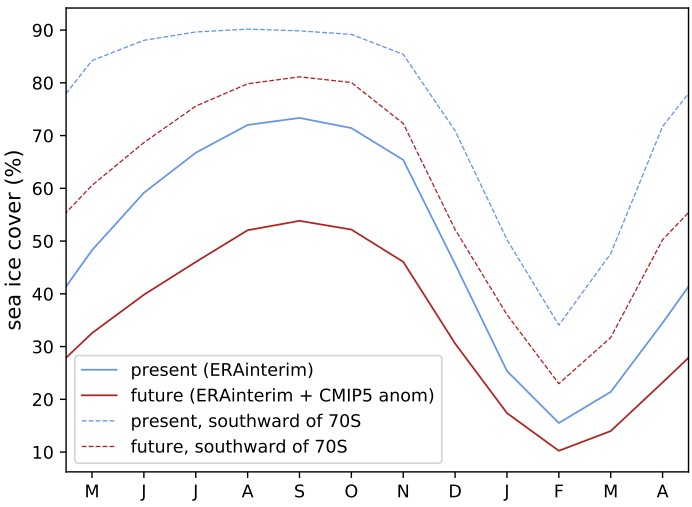

**Figure 2.** Mean seasonal cycle of sea-ice concentration over the oceanic part of the MAR domain (solid) and southward of 70°S (dashed), for the present-day (blue) and future (dark-red) simulations.

further. As expected from the radiative effects of greenhouse gases, the stratosphere tends to cool in response to increased anthropogenic emissions of greenhouse gases (e.g. Seidel et al., 2011). There is also a clear seasonal cycle in the lower stratosphere ($\sim$100 hPa), with future warming in spring and summer and cooling in the other seasons, which is related to seasonal effects of ozone recovery (Perlwitz et al., 2008). Specific humidity increases as the troposphere warms (Fig. 1b), as expected from the Clausius-Clapeyron relation.

## 3 Results

In this section, we present SMB and surface melt projections derived from ERA-interim and the CMIP5-MMM-RCP8.5 anomaly. We simply refer to the corresponding simulations as "present" and "future" in the following. We also investigate the causes for these changes and we discuss consequences for potential ice-shelf hydrofracturing and sea level rise.

### 3.1 Grounded ice-sheet SMB

The future SMB is increased by 30 to 40%, keeping a very similar pattern to present day (Fig. 3a,b), i.e. mostly controlled by the steep slopes and local topographic features near the ice-sheet margin. Considering the grounded part (which matters for sea level rise) of all the drainage basins from Getz to Abbot (boundaries indicated in Fig. 3a), SMB increases from 336 $\mathrm{Gt\,yr^{-1}}$ presently to 455 $\mathrm{Gt\,yr^{-1}}$ at the end of the 21$^{\mathrm{st}}$ century (Tab. 1). As previously reported by Ligtenberg et al. (2013), increasing snowfall explains most of the SMB changes. Projected sublimation slightly decreases in all basins, while rainfall slightly

increases, but both components remain two orders of magnitude smaller than snowfall. Runoff is projected to remain negligible over the grounded ice sheet in this sector.

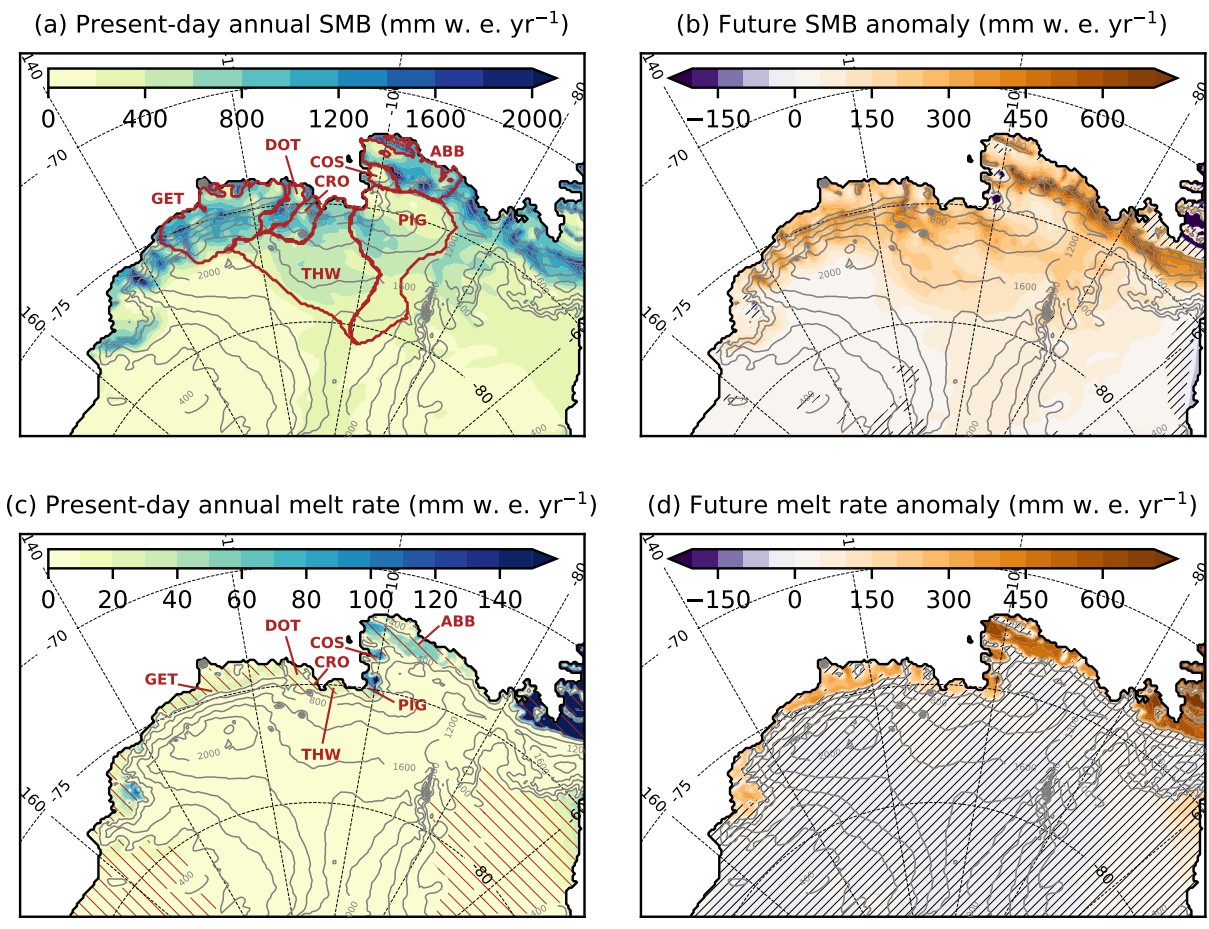

**Figure 3.** (a) Present-day annual mean SMB and (b) annual mean SMB anomaly (future minus present). (c) Present-day annual mean melt rate and (d) annual mean melt rate anomaly (future minus present). The dark red contours in the upper-left panel indicate individual glacial drainage basins (GET*z*, DOT*son*, CRO*sson*, THW*aites*, P*ine* I*sland* G*lacier*, COS*grove*, and ABB*ot*). Red hatching in the lower-left panels indicate ice shelves. Black narrow hatching on panels (b,d) indicate areas where the difference is not statistically significant (*t*-test on annual means, with a *p*-value of 0.05). The ice-sheet topography is shown in grey (contours every 400 m).

We now briefly analyse possible causes for increased SMB in a warmer climate. In the following, the relative increase $A$ in a variable $V$ (saturation water vapour pressure or snowfall) per degrees of warming is calculated by integrating:

$$
\begin{cases}
A & = \dfrac{1}{V}\dfrac{dV}{dT} \\
& = \dfrac{1}{T_2 - T_1}\ln\left(\dfrac{V_2}{V_1}\right)
\end{cases}
\tag{1}
$$

**Table 1.** SMB and its components over the grounded part of individual glacial drainage basins, for present day (regular) and future (bold). The results are here provided in $\mathrm{Gt\,yr^{-1}}$, i.e. integrated over the drainage basins (areas indicated in the second last row) to be directly convertible into a rate of sea level rise. SMB is the sum of snowfall and rainfall minus sublimation and runoff. Here we assume that the net production of surface liquid water is equivalent to runoff because of the significant slopes over the coastal grounded ice sheet. The bottom row shows the relative increase in snowfall per degree of air warming at 2 m above ground level (see eq. 1).

| SMB component $(\mathrm{Gt\,yr^{-1}})$ | Abbot | Cosgrove | Pine Island | Thwaites | Crosson | Dotson | Getz |
|---|---|---|---|---|---|---|---|
| SMB | 36.3 | 7.1 | 80.1 | 95.9 | 20.6 | 16.9 | 79.3 |
| | **50.8** | **10.0** | **110.2** | **129.1** | **28.4** | **22.9** | **103.5** |
| Snowfall | 37.0 | 7.3 | 82.0 | 95.6 | 21.0 | 17.3 | 81.0 |
| | **50.5** | **10.0** | **111.3** | **127.7** | **28.6** | **23.2** | **104.5** |
| Rainfall | 0.1 | 0.0 | 0.1 | 0.0 | 0.1 | 0.0 | 0.0 |
| | **0.7** | **0.1** | **0.3** | **0.1** | **0.2** | **0.1** | **0.2** |
| Sublimation | 0.7 | 0.2 | 2.0 | -0.2 | 0.5 | 0.4 | 1.7 |
| | **0.4** | **0.1** | **1.1** | **-1.3** | **0.4** | **0.3** | **1.2** |
| Runoff | 0.0 | 0.0 | 0.0 | 0.0 | 0.0 | 0.0 | 0.0 |
| | **0.1** | **0.0** | **0.3** | **0.0** | **0.0** | **0.0** | **0.0** |
| Area ($10^3$ km$^2$) | 30.0 | 8.8 | 186.3 | 192.4 | 23.5 | 16.2. | 90.0 |
| Snowfall rel. sensitivity ($\%\,^{\circ}\mathrm{C}^{-1}$) | +8.5 | +8.4 | +8.1 | +7.8 | +8.9 | +8.6 | +7.4 |

which has the advantage to give $A$ values that are relatively independent of the chosen $(T_2 - T_1)$ temperature interval given the approximate exponential relationship expected for the variables under consideration.

The saturation water vapour pressure increases with air temperature, at a rate of $7.1 \pm 0.1\,\%\,^{\circ}\mathrm{C}^{-1}$ in the 0-10°C range (Clausius-Clapeyron relation). In our simulations, near surface warming reaches 3.4 to 3.7°C for the various basins, which is very close to the RCP8.5 MMM global warming value (Collins et al., 2013). The corresponding increase in snowfall over the grounded ice sheet represents $+7.4$ to $+8.9\,\%\,^{\circ}\mathrm{C}^{-1}$ (bottom row of Tab. 1), which is higher than the theoretical Clausius-Clapeyron rate. This indicates that other factors may contribute to increasing snowfall in the Amundsen sector.

To further understand the mechanism for increased snowfall, we now consider projections for the four seasons separately (Fig. 4). The strongest increase in SMB occurs in MAM (followed by JJA), which corresponds to the season with largest

changes in sea-ice concentrations in the vicinity of the ice-sheet margin (see dashed lines in Fig. 2). While Clausius-Clapeyron refers to the maximum saturation water vapour pressure, we suggest that decreasing coastal sea-ice cover makes surface air masses closer to their saturation level, as previously suggested by Gallée (1996) and Kittel et al. (2018). This mechanism is also consistent with the modelling results of Wang et al. (2020) who find that precipitation over the coastal Amundsen

5   region mostly comes from evaporation occurring all the way from the Tropical Pacific to the Amundsen Sea. Another possible contributor to increased snowfall is the changing low-troposphere circulation, which shows a cyclonic anomaly in MAM, favouring humidity transport towards the ice sheet (Fig. 5). As warming at the height where precipitation is formed is relevant for Clausius-Clapeyron, a stronger warming far above the surface than in its vicinity could also contribute to explain this stronger sensitivity, although Fig. 1a suggests slightly stronger near-surface warming.

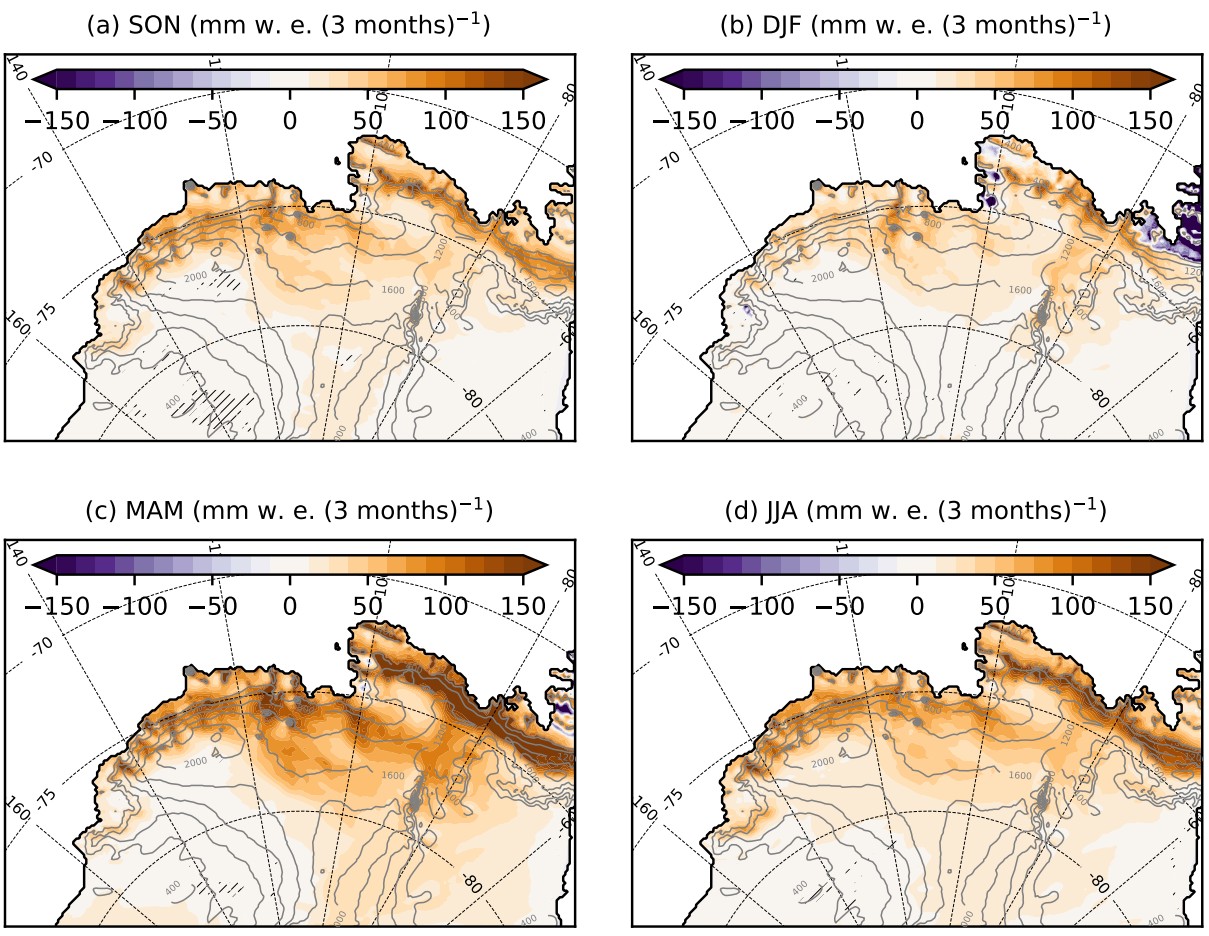

**Figure 4.** Changes in mean seasonal SMB (future minus present). Black narrow hatching indicate areas where the difference is not statistically significant ($t$-test on 3-month means, with a $p$-value of 0.05). The ice-sheet topography is shown in grey (contours every 400 m).

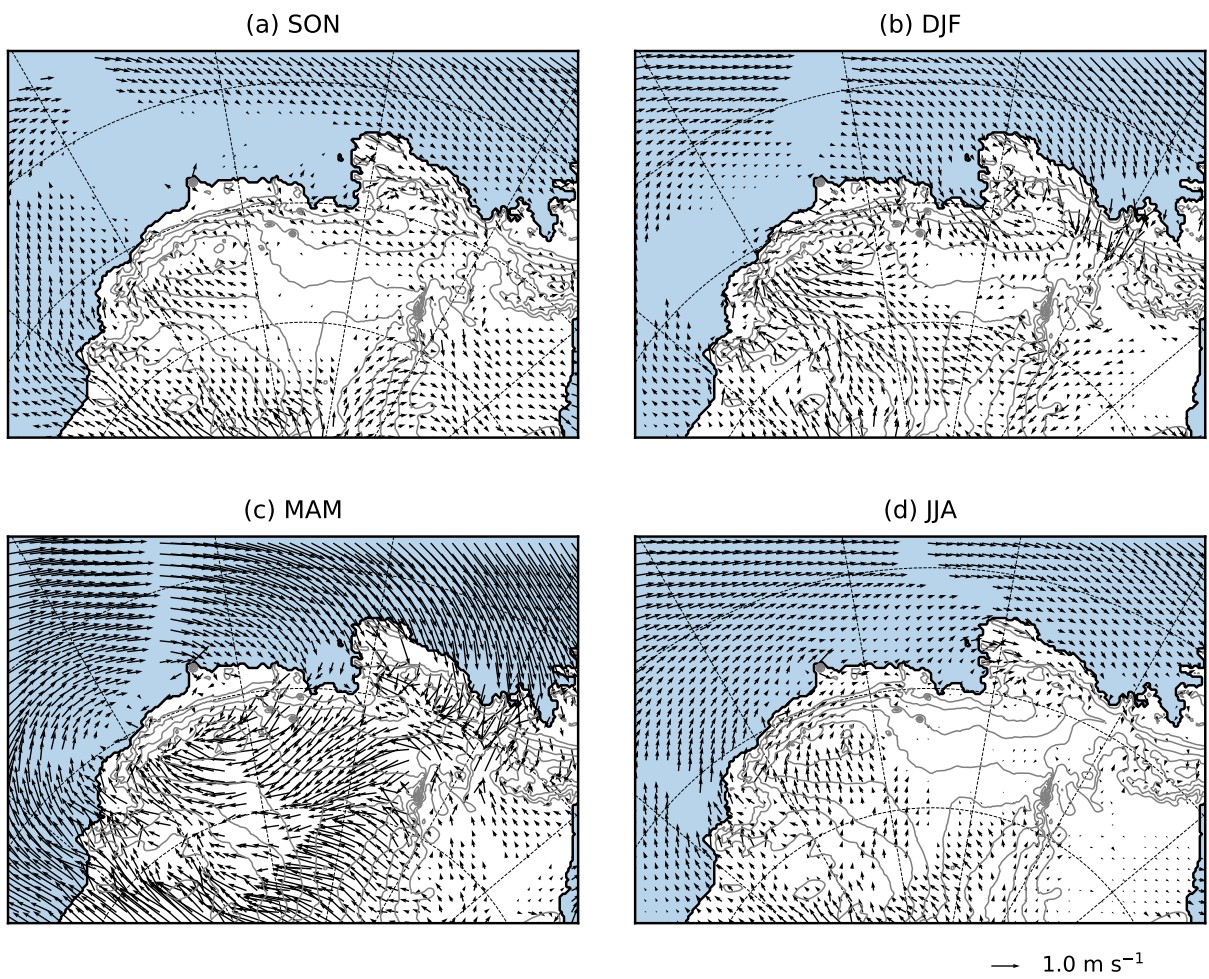

**Figure 5.** Changes in mean seasonal 10 m winds (future minus present). Vectors are not displayed at locations where the change in at least one of the wind components is not statistically significant ($t$-test on 3-month means, with a $p$-value of 0.05). The open ocean is in light blue and the ice-sheet topography is shown in grey (contours every 400 m).

## 3.2 Ice-shelf surface liquid water budget

We have shown that runoff plays no significant role in the simulated SMB over the grounded ice sheet and therefore on sea level projections. However, surface melt, rainfall and subsequent net production of surface liquid water may lead to ponding over ice shelves and trigger hydrofracturing. In this section, we therefore focus on liquid water budget projections over the seven major ice shelves from Getz to Abbot. In this paper, we do not investigate supra-glacial hydrology and hydrofracture mechanics in details, we simply consider the presence of net production of surface liquid water as an indicator of potential ice shelf collapse (i.e. necessary but not sufficient condition).

Surface melt rates averaged over the major individual ice shelves from Getz to Abbot are projected to increase by one order of magnitude, and melt occurrence is projected to increase from typically a week per year to 1-2 months per year (Tab. 2). As previously noticed by Kuipers Munneke et al. (2014) and Trusel et al. (2015), we find an exponential dependency of melt rates to 2 m air temperatures (not shown), with a much stronger dependency on temperature than SMB (Clausius-Clapeyron). In terms of seasonality, future melt rates are strongly increased in summer (DJF) over all the ice shelves, while Abbot, Cosgrove and Pine Island also experience significantly more melting in fall and spring (Fig. 6).

Rainfall is also projected to increase (Tab. 2), but represents a relatively small fraction of surface melt (less than 15% for all the ice shelves). Future surface melt and rainfall entirely refreeze in the firn for all the ice shelves from Getz to Thwaites, which leads to zero net surface liquid water production in the future. In contrast, Abbot, Cosgrove and Pine Island have a positive net surface liquid water production, although most surface melt and rainfall still refreeze in the firn.

The contrast between western (Getz to Thwaites) and eastern (Pine Island to Abbot) ice shelves can be explained by variations of the melt-to-snowfall ratio, which we now explain from simple considerations. As rainfall remains significantly weaker than melt rates, we neglect it in the following discussion, but more details on the theoretical role of rainfall are provided in Appendix B. First of all, if surface melt water percolates into snow layers that are below the freezing point, it partly refreezes, which releases latent heat and warms the snow layers. Therefore, the melt-to-snowfall ratio must typically exceed a few hundredth to bring the snow to $0°C$ and allow the existence of liquid water in snow. To have a net production of surface liquid water, melt rates also need to be sufficiently high to significantly deplete air in snow. Based on a simple model, Pfeffer et al. (1991) estimated that surface melt would lead to snow-air depletion for melt-to-snowfall ratios greater than approximately 0.7, considering fresh-snow and close-off densities of 300 and 830 $\mathrm{kg\,m^{-3}}$ respectively (see Appendix B). This shows that in the presence of relatively fresh snow, large melt-to-snowfall ratios are needed to have a net production of surface liquid water because large melt rates are needed to fill the porosity brought by large snowfall. In contrast, small quantities of melt or rain water are buried in large snowfall and likely refreeze.

Going back to our simulations, we note the importance of the melt-to-snowfall ratio for the net production of surface liquid water simulated by MAR over the ice shelves, with episodic production for annual melt-to-snowfall ratios as low as 0.25, and a highly probable production for annual melt-to-snowfall ratios greater than ∼0.85. (Fig. 7). The ratio allowing the production of surface liquid water exhibits some variability due to varying snow characteristics and a more complex firn model than in

**Table 2.** Components of the surface liquid water budget and snowfall over individual ice shelves, for present day (regular) and future (bold). "Net Liquid" is the sum of Melting plus Rainfall minus Refreezing minus the liquid water retained in the firn without refreezing (zero in our case, not shown). Here we present average values in $\mathrm{mm\,w.\,e.\,yr^{-1}}$ (mm water equivalent per year, i.e. $\mathrm{kg\,m^{-2}\,yr^{-1}}$), which is thought to be more meaningful than integrated values in terms of hydrofracture potential. The middle row indicates the melt to snowfall ratio. The bottom rows indicate the number of rain days (threshold of $1\,\mathrm{mm\,w.\,e.\,day^{-1}}$) and the number of melt days per year (threshold of $3\,\mathrm{mm\,w.\,e.\,day^{-1}}$ as in Donat-Magnin et al., 2020).

| Component ($\mathrm{mm\,w.\,e.\,yr^{-1}}$) | Abbot | Cosgrove | Pine Island | Thwaites | Crosson | Dotson | Getz |
|---|---|---|---|---|---|---|---|
| Melting | 54 | 80 | 79 | 26 | 17 | 21 | 22 |
|  | **577** | **588** | **455** | **244** | **183** | **292** | **333** |
| Refreezing | 60 | 83 | 85 | 29 | 20 | 24 | 25 |
|  | **613** | **462** | **372** | **268** | **201** | **310** | **348** |
| Rainfall | 6 | 3 | 6 | 3 | 3 | 3 | 2 |
|  | **77** | **27** | **33** | **25** | **18** | **18** | **17** |
| Net Liquid | 0 | 0 | 0 | 0 | 0 | 0 | 0 |
|  | **40** | **153** | **116** | **0** | **0** | **0** | **1** |
| Snowfall | 790 | 290 | 407 | 809 | 1055 | 669 | 786 |
|  | **943** | **372** | **521** | **989** | **1339** | **830** | **978** |
| Melt / Snowfall | 0.07 | 0.28 | 0.19 | 0.03 | 0.02 | 0.03 | 0.03 |
|  | **0.61** | **1.58** | **0.87** | **0.25** | **0.13** | **0.35** | **0.34** |
| Nb rain days /yr | 1.0 | 0.6 | 1.0 | 0.5 | 0.4 | 0.3 | 0.4 |
|  | **14.0** | **5.7** | **5.8** | **4.2** | **3.1** | **2.9** | **3.1** |
| Nb melt days /yr | 7.8 | 10.7 | 10.2 | 3.5 | 2.3 | 2.9 | 3.1 |
|  | **65.3** | **63.2** | **46.6** | **31.0** | **25.3** | **38.0** | **42.7** |

Pfeffer et al. (1991), but on average, surface liquid water becomes more likely than not for melt-to-snowfall ratios greater than 0.85.

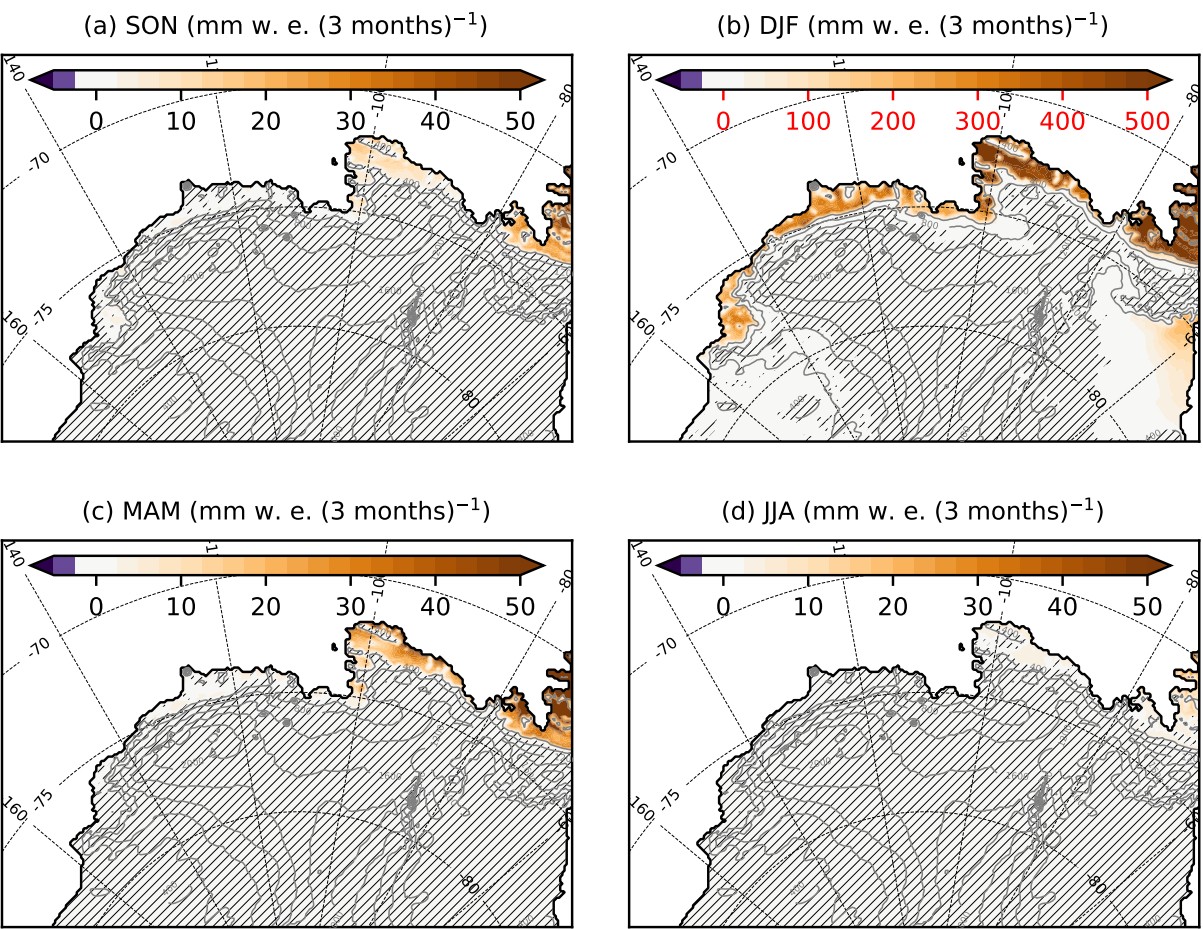

**Figure 6.** Changes in seasonal mean melt rates (future minus present). The colour-bar labels in panel (b) are shown in red to emphasise the different range compared to other panels. Black hatching indicates areas where the difference is not statistically significant ($t$-test on 3-month means, with a $p$-value of 0.05). The ice-sheet topography is shown in grey (contours every 400 m).

The existence of such a threshold explains the variations in liquid water production across the ice shelves (middle row of Tab. 2): Abbot, Cosgrove and Pine Island have relatively high future melt rates ($\sim 500$ mm w.e. yr$^{-1}$) but Abbot receives much higher snowfall, which explains that surface melt produces less surface liquid water than over Cosgrove and Pine Island; the four other ice shelves experience both relatively high snowfall and weak melt rates, which explains the absence of net production of surface liquid water in a warmer climate. Concerning Pine Island, it should be noted that high melt rates are concentrated on its north-eastern flank (Figs. 3,6), so potential hydrofracturing may be limited to that part, which is not the most important in terms of ice sheet dynamics and instability (e.g. Favier et al., 2014).

We now briefly analyse the causes for increased melting in a warmer climate. All along the future melting season, less energy is lost by the ice-shelf surface through net longwave radiation (Fig. 8b), which is a consequence of higher downward

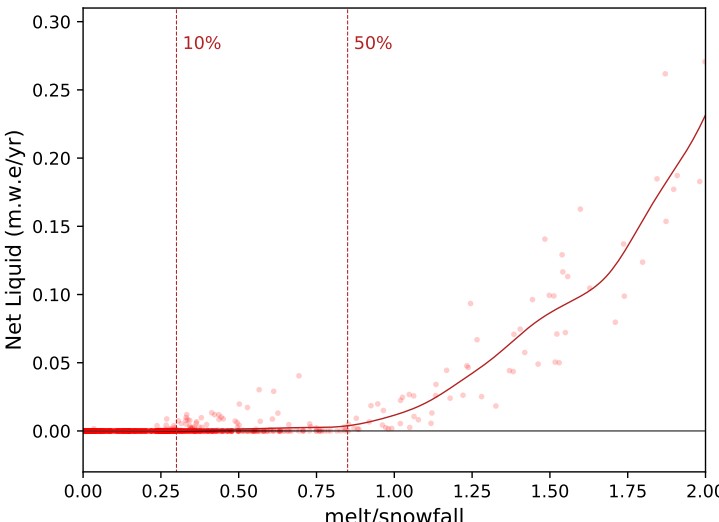

**Figure 7.** Net production of surface liquid water vs melt-to-snowfall ratio in the future simulation (calculated from climatological means). Each circle represents the climatological annual mean at a grid point within the seven glacial drainage basins. The solid curve is a Gaussian kernel density estimate with a standard deviation of 0.1 melt-to-snowfall ratio. The vertical dashed lines indicate the limit above which more than 10% and 50% of the points experience a net production greater than $1 \, \mathrm{mm\,w.\,e.\,yr^{-1}}$.

longwave radiation, as expected in the presence of higher specific humidity, only partly compensated by higher upward long-wave radiation emitted by a warmer snow surface in the future (Fig. 8a,b). In the future, more energy is also received by the snow surface through shortwave radiative fluxes over the melting season (Fig. 8c), which is explained by a melt-albedo feedback, i.e. a decreased ice-shelf albedo as a result of more melting (Fig. 8e). These changes are partly compensated by less shortwave radiation received by the snow surface (negative anomaly of the downward component in Fig. 8a), which is explained by a moderate increase in summer cloudiness (not shown) in the future. Changes in sensible and latent heat fluxes are less important than changes in radiative forcing, but they compensate a part of the increased net longwave and shortwave radiations (Fig. 8a,d). This may be related to a thicker planetary boundary layer in the future (Fig. 8f), i.e. reduced near-surface temperature and humidity vertical gradients, similar to the difference between summer and winter (Fig. 8d,f).

## 4 Discussion

We first discuss the possibility to extrapolate our results to other climate perturbations. Then, we discuss some limitations of our modelling and methodological approaches and their impacts on our projections.

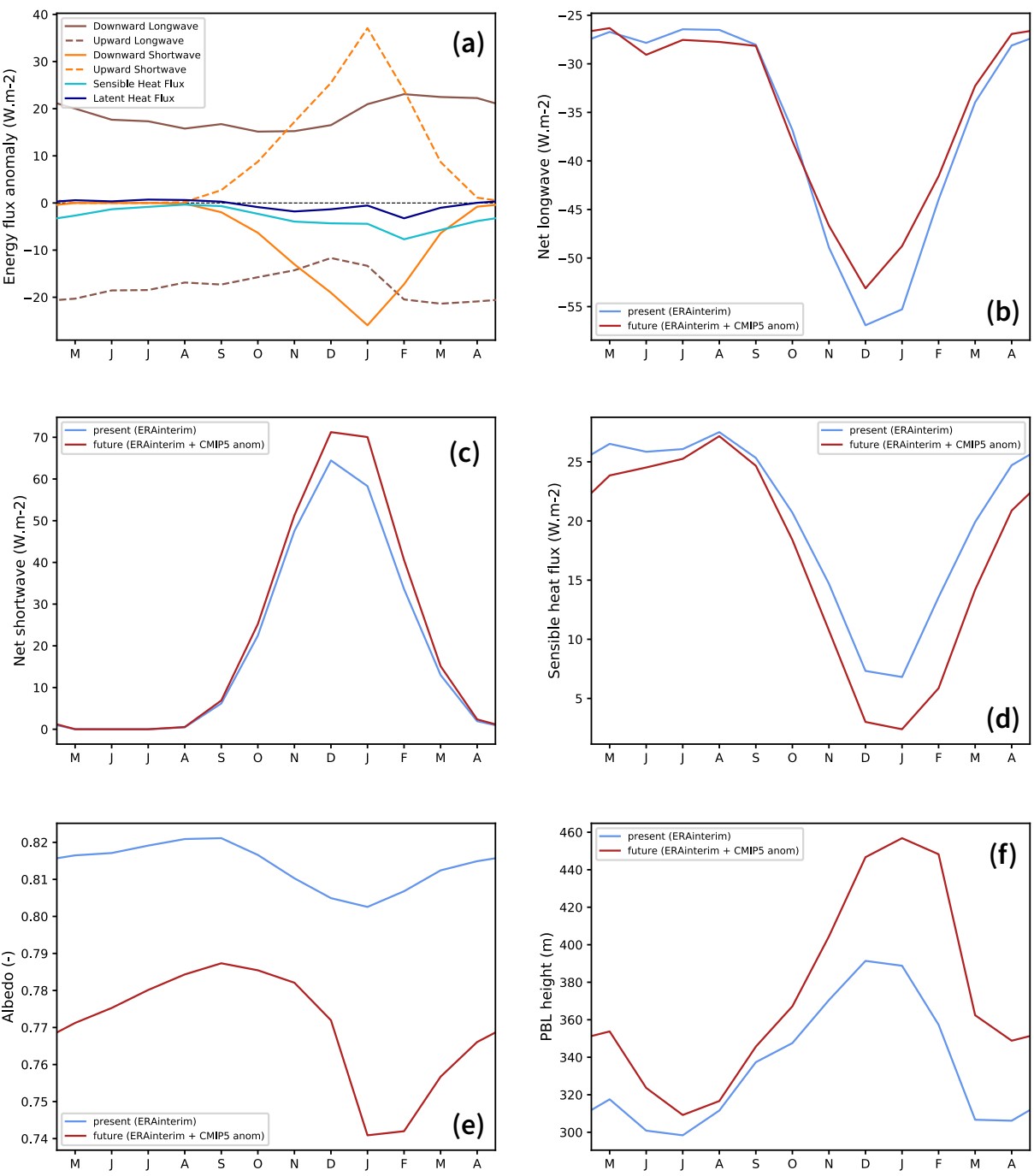

**Figure 8.** (a) Seasonal cycle of the anomaly (future minus present) in energy fluxes received by the ice-shelf surface (averaged ocer the seven major ice shelves from Getz to Abbot; positive if received by the surface). Present and future (b) net longwave radiative, (c) net shortwave radiative, and (d) sensible heat fluxes received by the ice-shelf surface. Mean present and future (e) albedo and (f) planetary-boundary-layer (PBL) height averaged over the seven major ice shelves from Getz to Abbot. The PBL height is calculated online in MAR from the vertical profile of horizontal turbulent kinetic energy.

## 4.1 Extrapolation to other climate perturbations

While CMIP5-MMM-RCP8.5 at the end of the 21[st] century is meaningful, it is also interesting to estimate the likelihood of net production of surface liquid water over the ice shelves further in the future or following alternative emission scenarios. To do so, we evaluate the melt-to-snowfall ratio for a given additional warming or cooling, assuming that snowfall (SNF) and melt rates (MLT) evolve following simple relationships with temperature. Such relationships can be obtained from the literature. The snowfall (SNF) dependency to temperature can be obtained by the Magnus empirical fit of the Clausisus-Clapeyron relationship (Koutsoyiannis, 2012), here further simplified by linearizing the term of the exponential around 0°C . The melt rate (MLT) has also an exponential dependency to near surface temperature, with an empirical expression derived by Trusel et al. (2015) from a numerical model applied to the entire Antarctic ice sheet. For a given ice shelf ($is$), this yields:

$$
\left\{
\begin{aligned}
\mathrm{SNF}_{is}(\Delta T) &= \mathrm{SNF}_{is,\mathrm{p}} \exp\left(\frac{17.625(\Delta T - \Delta T_{\mathrm{p}})}{243.04}\right) &= \mathrm{SNF}_{is,\mathrm{p}} \exp\left(0.072(\Delta T - \Delta T_{\mathrm{p}})\right) \\
\\
\mathrm{MLT}_{is}(\Delta T) &= \mathrm{MLT}_{is,\mathrm{p}} \exp\left(0.456(\Delta T - \Delta T_{\mathrm{p}})\right) \\
\\
R_{is}(\Delta T) &= \frac{\mathrm{MLT}_{is}(\Delta T)}{\mathrm{SNF}_{is}(\Delta T)} &= R_{is,\mathrm{p}} \exp\left(0.384(\Delta T - \Delta T_{\mathrm{p}})\right)
\end{aligned}
\right.
\tag{2}
$$

where $\Delta T$ represents warming with respect to present-day (1989-2009) and $\Delta T_p$ is the CMIP5-MMM-RCP8.5 warming analysed in this study (2080-2100 minus 1989-2009). The two first lines of (2) are obtained by using the simulated future values on individual ice shelves at $\Delta T = \Delta T_{\mathrm{p}}$, i.e. $\mathrm{SNF}_{is,\mathrm{p}}$ and $\mathrm{MLT}_{is,\mathrm{p}}$. The third line gives the melt-to-snowfall ratio of a given ice shelf ($R_{is}$).

While the expressions in (2) have the advantage to be theoretically valid for general Antarctic conditions, local fits based on our simulations are also meaningful. Another expression can be derived assuming the same exponential form as (2) but with a coefficient in the exponential calculated as the average of the seven values calculated for individual ice shelves:

$$
R_{is} = R_{is,\mathrm{p}} \exp\left(0.577(\Delta T - \Delta T_{\mathrm{p}})\right)
\tag{3}
$$

This second method gives a stronger sensitivity to warming than (2). Recalculating an exponential fit for melt rates in a similar way as Trusel et al. (2015) also gives a slightly stronger sensitivity ($\Delta\mathrm{MLT} = 853 \exp\left(0.55\Delta T\right)$ in $\mathrm{mm\,w.\,e.\,yr}^{-1}$), which can be a specificity of either the Amundsen region or our model configuration.

The extrapolations corresponding to (2) and (3) are shown in Fig. 9. Both expressions are kept in order to estimate the uncertainty. In terms of scenarios, these extrapolations suggest that no ice shelf would experience a net production of surface liquid water over the 21[st] century under the RCP2.6 scenario, and only Cosgrove would experience this before 2100 under the RCP4.5 scenario. Under the RCP8.5 scenario, our extrapolations suggest that Cosgrove would likely experience a net production of surface liquid water by 2050, followed by Pine Island before 2100 and Abbot near 2100. Surface liquid water would also be produced in excess over the remaining ice shelves from the middle of the 22[nd] century, except Crosson that could remain relatively free of surface liquid water until the 23[rd] century. Due to the generally higher climate sensitivity of CMIP6

models (e.g., Zelinka et al., 2020), extrapolations for the SSP126, SSP245 and SSP585 scenarios indicate that more ice shelves could experience a net production of surface liquid water before the end of the 21st century.

The increasing proportion of liquid precipitation in a warmer climate is neglected in the above equations although it may contribute to the production of surface liquid water. Rainfall remains significantly weaker than melt rates in our RCP8.5 projections (at most 15% of melt rates in Table 2) and its capacity to deplete snow/firn air is weaker than melt rates (see Appendix B), but accounting for increasing rainfall might slightly advance the onset of net surface liquid water production late in the 22nd century and in the 23rd century. In MAR simulations driven by CMIP6 models of high climate sensitivity, Kittel et al. (2020) (their Tab. 1) found that rainfall could become as large as snowfall over the Antarctic ice shelves by the end of the 21st century, but corresponding melt rates would be 7 to 8 times larger than rainfall, indicating that the net production of surface liquid water remains mostly related to melt rates in conditions warmer than in our MAR projections.

These results are difficult to compare precisely to previous studies because different metrics and scenarios were used. Based on the CMIP3 HadCM3 model under the A1B scenario (similar global warming as CMIP5-MMM-RCP8.5 in 2100), Kuipers Munneke et al. (2014) found that 50% of the present-day firn air thickness would be depleted by ∼2130 for Cosgrove and ∼2085 for Abbot. Assuming that this corresponds to our 0.85 melt-to-snowfall threshold, we rather find ∼2055 for Cosgrove and ∼2100 for Abbot. Besides, Kuipers Munneke et al. (2014) found little firn air depletion by 2200 under A1B for all the ice shelves from Thwaites to Getz, while we find that firn air could be fully depleted at Getz and Dotson before 2200. The generally later full-depletion dates in their simulations could be related to the ∼-1.5°C present-day bias in the regional atmospheric simulations used to drive their firn model. To estimate the likelihood of ice shelf collapse in future scenarios, Trusel et al. (2015) used melt-rate thresholds (based on pre-collapse observations at Larsen B). They found that only Abbot could reach this threshold by 2100 and only under the RCP85 scenario, but given the large snowfall spatial variability around Antarctica and across the Amundsen region, we believe that the melt-to-snowfall ratio would be a better indicator of potential ice-shelf collapse than a uniform melt-rate threshold as used in Nowicki et al. (2020) and Seroussi et al. (2020).

## 4.2 Modelling and methodological limitations

We now assess the ability of our projection method to capture the future climatology in a similar way as Yoshikane et al. (2012), i.e. running a perfect-model test (i.e. assuming that the future is perfectly known by considering that a given projection is true). To do so, we now consider a single model, namely ACCESS-1.3 (Bi et al., 2013; Lewis and Karoly, 2014), which reproduces remarkably well the present-day climate over Antarctica (Agosta et al., 2015; Naughten et al., 2018; Barthel et al., 2020). We first run MAR forced by ACCESS-1.3 over 1989-2009 and 2080-2100 under the RCP8.5 scenario, and we consider 2080-2100 from this run as the *true future*. Then, we calculate the seasonal climatological anomaly and add it to the present-day interannual forcing, i.e. following the methodology described in the previous section but using present-day ACCESS-1.3 and its future anomalies instead of ERA-interim and the CMIP5 MMM anomaly. The future based on the absolute anomaly method is referred to as *projected future* in this section, and it is compared to the *true future* (from the direct downscaling of ACCESS-1.3). The fidelity of our projection method is assessed by comparing the difference between the *projected future* and the *true future* (i.e., the projection bias) to the true climate change signal (*true future* minus present). We can see that our

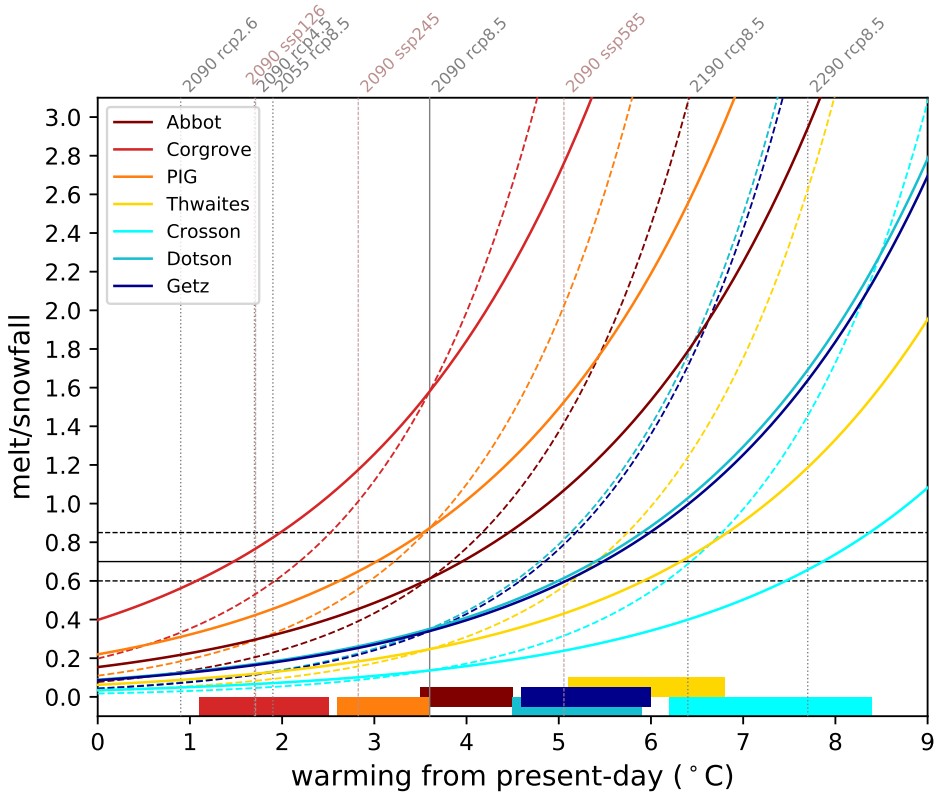

**Figure 9.** Extrapolated melt-to-snowfall ratio as a function of warming with respect to present-day (solid lines correspond to Eq. 2 and dashed lines to Eq. 3). The values obtained through our simulations correspond to the intersections with the vertical solid grey line. The vertical dashed grey lines represent warming at other dates (the dates indicated above the lines are the central years of 20-year averages) and under alternative scenarios (RCP2.6 and RCP4.5). This warming is derived from Collins et al. (2013, their Tab. 12.2), assuming that the regional warming remains equal to global warming (supported by our results as well as Collins et al., 2013). The corresponding warming for the CMIP6 multi-model mean (see list in Appendix A) under scenarios SSP126, SSP245 and SSP585 are indicated as vertical thin dashed rosy-brown lines. The black horizontal lines indicate three indicative thresholds for a net production of surface liquid water: the future 0.60 ratio simulated at Abbot in 2080-2100 (which is the minimum ratio for which we detect significant production of surface liquid water), the 0.70 ratio estimated by Pfeffer et al. (1991), and the 0.85 ratio for which more than 50% of the grid points experience a net production of surface liquid water (Fig. 7). The warming range for which the extrapolations cross the 0.60 and 0.85 thresholds are indicated by the horizontal color bars at the bottom.

iterative sea-ice correction (see section 2) is effective, reducing the SIC projection bias from 14% to 0.3% of the climate-change anomaly in SON, and from 40% to 20% of the climate-change anomaly in DJF (Fig. 10a).

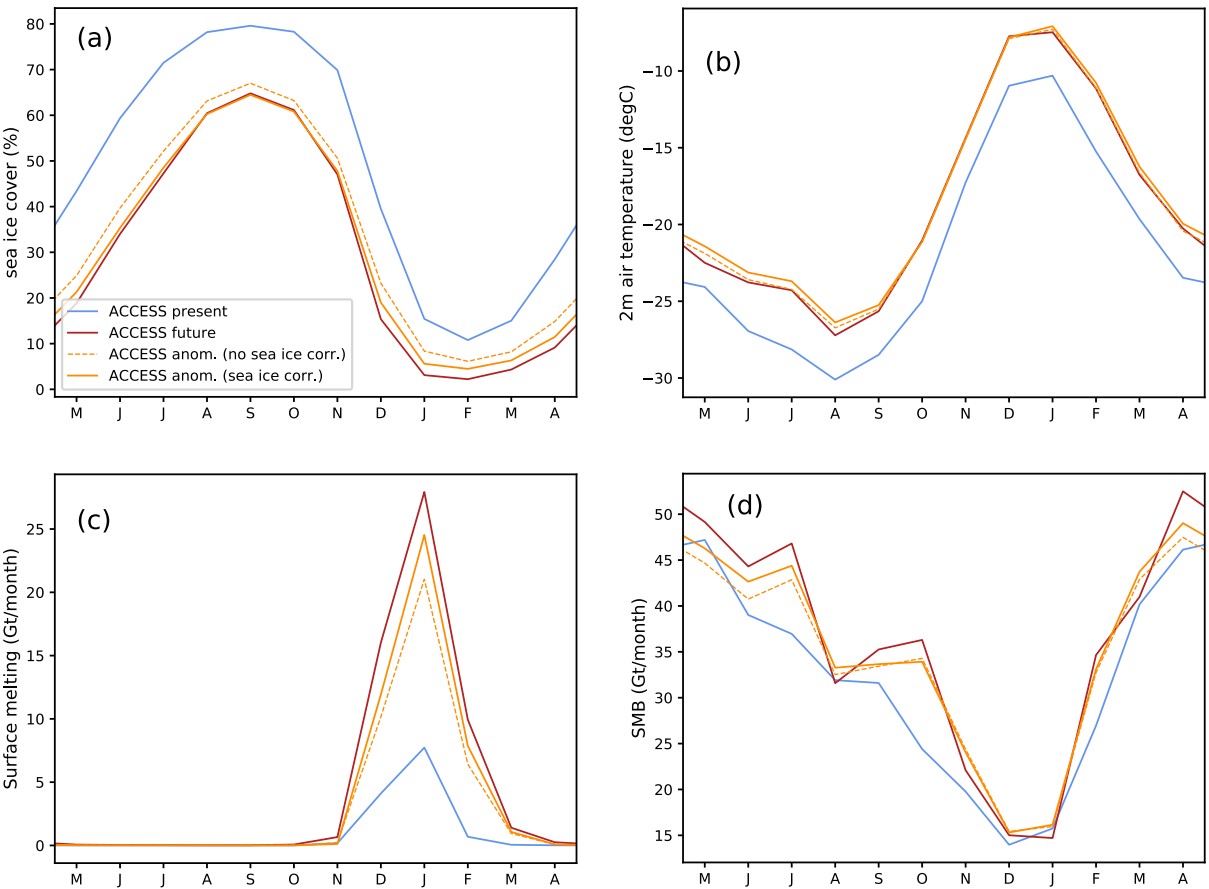

**Figure 10.** Mean 21-year seasonal cycle of: (a) domain-averaged sea-ice concentration, (b) 2 m air temperature, (c) surface melt and (d) SMB, with (b-d) integrated over the seven glacial drainage basins from Getz to Abbot (including both ice shelf and grounded ice). The blue and dark-red lines correspond to the present and *true future* based on ACCESS-1.3, respectively. The orange lines represent the results of the *projected future*, applied with (solid) and without (dashed) sea-ice iterative correction (see section 2).

Over the ice sheet, the near-surface projection biases are 0.6°C in JJA and 0.2°C in DJF, which is relatively small compared to a warming signal of 3.5°C and 3.0°C for these two seasons respectively (Fig. 10b). Looking at the peak melt rate in January (Fig. 10c), we find that the projection bias represents 17% of the climate-change signal, vs 34% if no iterative method is used for sea ice. The annual SMB projection bias represents 16% of the projected increase, vs 32% if no iterative method is used for sea ice (Fig. 10d). In terms of spatial pattern, the climate change signal remains significantly larger than the projection bias at most locations (Fig. 11a,b). The melt projection bias is positive at most melting locations, with a bias consistently smaller than the climate change signal (Fig. 11c,d).

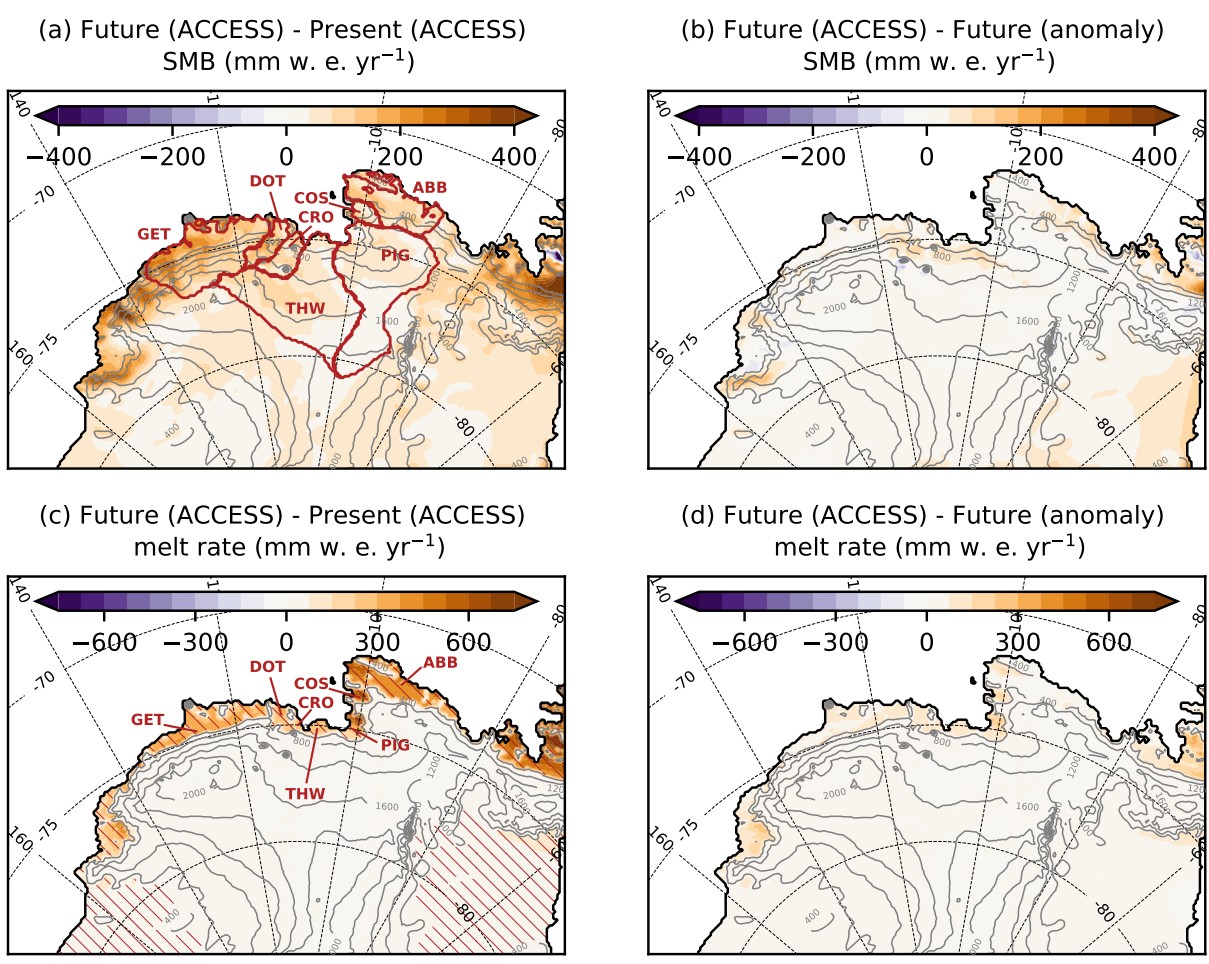

**Figure 11.** (a) SMB anomaly (2080-2100 minus 1989-2009) in ACCESS-1.3 directly downscaled by MAR (i.e. *true* changes). (b) Difference between the *true future*, from the direct downscaling of ACCESS-1.3, and the *projected future*, from the anomaly method. (c-d) Same as (a-b) but for melt rates instead of SMB. The dark red contours in the upper-left panel indicate individual glacial drainage basins (GET*z*, DOT*son*, CRO*sson*, THW*aites*, P*ine* I*sland* G*lacier*, COS*grove*, and ABB*ot*). Red hatching in the lower panels indicate ice shelves. The ice-sheet topography is shown in grey (contours every 400 m).

To summarise our assessment of our projection method, it has the advantage to start from a present-day state that is not affected by present-day biases in CMIP5 models and to be applicable to a multi-model-mean projection, which is expected to remove a part of the CMIP5 model biases. The counterpart of these advantages are biases in the projection itself. These biases are estimated to remain below 20% based on our perfect-model approach. A part of these biases may be related to the imperfect method used to apply the sea-ice anomaly. Using iterative absolute anomalies typically removes half of the projection biases compared to a simple absolute anomaly, but the bias is not completely removed in summer, and more iterations or a refined method may be needed in our approach. Alternative approaches to build future sea-ice concentrations were proposed by Beaumet et al. (2019b), and some of them may be more effective at removing projection biases, although their approaches produced biases of similar magnitude as our iterative absolute anomaly method (their Fig. 5). Another possible cause for our projection biases is the fact that we assume unchanged interannual variability with respect to the mean in the *projected future*, while the *true future* experiences a different variability. Changes in interannual variability or extreme events may indeed affect non-linear processes (e.g., melt rates vary exponentially with temperatures) even if the mean is the same in the *true future* and the *projected future*. Notwithstanding these limitations, we consider that our methodology has some advantages and should be used for projections together with other existing methods.

We now discuss the consequences of the aforementioned model and methodological biases for future surface liquid water production and potential hydrofracturing. Our projection method produces an underestimation of both snowfall and melt rates in the future by 16-17%. Adding these errors to both snowfall and melting values in Tab. 2 would keep the melt-to-snowfall ratio unchanged. As such, the projection bias is not expected to change the list of ice shelves experiencing future production of surface liquid water. Besides, the melt rates and snowfall produced by MAR in this configuration were shown to be biased by typically $-20\%$ and $+20\%$ respectively (Donat-Magnin et al., 2020), although observational melt-rate products are also highly uncertain (Datta et al., 2019). Increasing melt rates in Tab. 2 by 20% and reducing snowfall by 20% changes the melt-to-snowfall ratios, bringing Abbot's to 0.92, i.e. well above the critical threshold. This again shows the high sensitivity of projected surface liquid water production at the surface of Abbot. Nonetheless, Thwaites (ratio changed to 0.37), Crosson (0.20), Dotson (0.53) and Getz (0.51) keep a low probability to experience a net production of surface liquid water even accounting for possible model biases. These estimates suggest that the absence of liquid water at the surface of Thwaites, Crosson, Dotson and Getz in 2100 under CMIP5-MMM-RCP8.5 conditions is a robust feature.

We now discuss another critical aspect of firn modelling, which is the spin-up duration. Our approach has consisted of running a present and a future 30-year snapshot, which means that the future firn has not experienced transient changes throughout the 21$^{st}$ century. Instead, we have run a 12-year spin up under future conditions for every simulated year of the future experiment (the years are run in parallel). We now consider surface liquid water produced in DJF 1998 with climate anomalies on top, which is the summer with highest melt rates in our projection and is preceded by a decade of relatively high melt rates (Donat-Magnin et al., 2020). We consider that the spin up duration is sufficiently long if the net production of surface liquid water in DJF-1998 reaches a steady state for spin-up durations shorter than 12 years. Whatever the spin-up duration, there is no significant net production at the surface of Getz, Dotson, Crosson and Thwaites (Fig. 12), which is expected due to the low melt-to-snowfall ratio (see previous section). For Pine Island and Cosgrove, an approximate steady state seems to be reached

after 6-7 years, although the net production at Cosgrove still experiences fluctuations of ±10%. In contrast, the net production over Abbot is still drifting after 12 years of spin up. This is likely related to the relatively weak but non-zero net production associated with a melt-to-snowfall ratio close to the critical threshold, which probably means that the firn is still slowly filling up after 12 years. Expanding the spin-up duration much further under constant 2080-2100 conditions would not make a lot of sense as earlier conditions were less affected by climate warming. We suggest that simulating the transient 21$^{st}$ century may be needed to set up the future firn properties of Abbot, and our results concerning this ice shelf have therefore to be considered carefully.

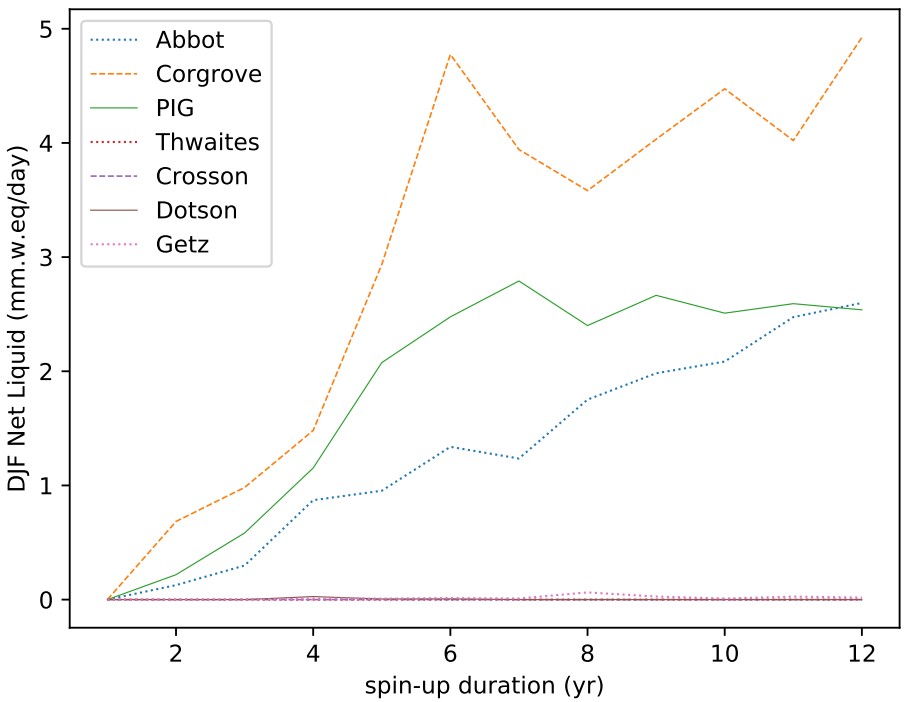

**Figure 12.** Net production of surface liquid water over individual ice shelves in DJF 1998 for various spin-up durations, for GET*z*, DOT*son*, CRO*sson*, THW*aites*, P*ine* I*sland* G*lacier*, COS*grove*, and ABB*ot*.

## 5 Conclusion

In this study, we have presented future projections of SMB and surface melt at the end of the 21$^{st}$ century under the RCP8.5 scenario, based on the MAR regional atmospheric model at 10 km resolution. The climate change anomaly is calculated from the seasonal climatology of a CMIP5 multi-model mean, and added to the ERA-interim reanalysis which is used for present-

day boundary conditions. The use of an anomaly has the advantage to start from a present state with small biases compared to observations, and is expected to reduce future biases as most CMIP5 biases were shown to be stationary. Besides, the use of a multi-model mean is expected to cancel the biases that are not common to a majority of models. An important caveat of this method is that we assume unchanged interannual variability with respect to the mean. A perfect-model test indicates that our approach captures future changes in most variables, despite an underestimation of SMB and melt rate changes by 17% on average.

Considering the drainage basins of the seven major ice shelves from Getz to Abbot, and only for the grounded parts of the ice sheet, we find that SMB increases from $336 \, \mathrm{Gt \, yr^{-1}}$ to $455 \, \mathrm{Gt \, yr^{-1}}$ throughout the 21$^{st}$ century, which would reduce the global sea level changing rate by $0.33 \, \mathrm{mm \, yr^{-1}}$. Even in the future climate, SMB over the grounded ice sheet remains nearly equivalent to snowfall in this region. Snowfall increases by 7.4 to 8.9% per °C of near-surface air warming, which is similar to global warming in this region. This sensitivity is slightly larger than previous estimates for the whole ice sheet (Palerme et al., 2017; Lenaerts et al., 2016; Ligtenberg et al., 2013, and references therein), and larger than predicted by Clausius-Clapeyron (increase in saturation vapour pressure by 7 to $7.5\% \, \mathrm{°C^{-1}}$). This may be explained by a decreased sea-ice cover along the ice-sheet margin, which helps near-surface air masses to reach their water vapour saturation. Changes in local circulation in autumn, and associated advection of marine air, may also favour higher SMB in the future.

Then, we have analysed future surface melt and the liquid water budget at the surface of ice shelves because they can lead to hydrofracturing and ice-shelf collapse. At the surface of the seven major ice shelves between Getz and Abbot, future melt rates are increased by an order of magnitude compared to present day, and the average number of melt days per year in the future exceeds 30 for most ice shelves. However, most melt water refreezes in the firn, even in the future run, as previously found by Kuipers Munneke et al. (2014) and Ligtenberg et al. (2013). Hence, significant amounts of surface liquid water (produced after warming of the snowpack and depletion of the firn air content by melt water) are only found over Abbot, Cosgrove and Pine Island ice shelves at the end of the 21$^{st}$ century. All the ice shelves from Thwaites to Getz are projected to remain nearly free of surface liquid water throughout the 21$^{st}$ century. The melt-to-snowfall ratio explains regional contrasts in our projections, and the net production of surface liquid water becomes significant if this ratio exceeds 0.60 to 0.85. Based on the melt and snowfall dependencies to near-surface warming, we have extrapolated our projections further in time and for other scenarios. Although uncertain, this suggests that most ice shelves could remain free of surface liquid water by 2100 under RCP2.6 and RCP4.5, to the exception of Cosgrove. Under RCP8.5, the ice shelves from Thwaites to Getz may only experience net production of surface liquid water in the second half of the 22$^{nd}$ century, and possibly the 23$^{rd}$ century in the case of Crosson. These results suggest that for Getz, Dotson, Crosson and Thwaites, ice-shelf collapse is unlikely to be triggered by hydrofracturing before the 22$^{nd}$ century. Nonetheless, it does not mean that these ice shelves will not collapse through other mechanisms, in particular ocean-induced basal melting, as observed and projected for Thwaites (e.g. Milillo et al., 2019; Yu et al., 2019), with positive feedbacks to ice damage and ice discharge (Lhermitte et al., 2020).

## Appendix A: List of CMIP5 models

The following ensemble of 33 CMIP5 models has been used in this study: ACCESS1-0, ACCESS1-3, BNU-ESM, CCSM4, CESM1-BGC, CESM1-CAM5, CESM1-WACCM, CMCC-CESM, CMCC-CMS, CMCC-CM, CNRM-CM5, CSIRO-Mk3-6-0, CanESM2, FGOALS-g2, FIO-ESM, GFDL-CM3, GFDL-ESM2G, GFDL-ESM2M, HadGEM2-CC, HadGEM2-ES, IPSL-CM5A-LR, IPSL-CM5A-MR, IPSL-CM5B-LR, MIROC-ESM-CHEM, MIROC-ESM, MIROC5, MPI-ESM-LR, MPI-ESM-MR, MRI-CGCM3, NorESM1-ME, NorESM1-M, bcc-csm1-1, inmcm4. We take the first available ensemble member for each model (i.e. "r1i1p1" or "r2i1p1" if not available).

The following ensemble of 33 CMIP6 models has been used to calculate the global warming values corresponding to scenarios ssp126, ssp245 and ssp585 in Fig. 9: ACCESS-CM2, ACCESS-ESM1-5, AWI-CM-1-1-MR, BCC-CSM2-MR, CAMS-CSM1-0, CanESM5, CESM2, CESM2-WACCM, CNRM-CM6-1-HR, CNRM-CM6-1, CNRM-ESM2-1, EC-Earth3, EC-Earth3-Veg, FGOALS-f3-L, FGOALS-g3, FIO-ESM-2-0, GFDL-ESM4, GISS-E2-1-G, HadGEM3-GC31-LL, INM-CM4-8, INM-CM5-0, IPSL-CM6A-LR, KACE-1-0-G, MCM-UA-1-0, MIROC6, MIROC-ES2L, MPI-ESM1-2-HR, MPI-ESM1-2-LR, MRI-ESM2-0, NESM3, NorESM2-LM, NorESM2-MM, UKESM1-0-LL.

## Appendix B: Air depletion by snowfall and rainfall

Here we extend the approach of Pfeffer et al. (1991) to give a very simple description of the effects of both rainfall and surface melt on snow saturation by liquid water.

We consider a snowfall rate SNF (in $\mathrm{kg\,m^{-2}\,s^{-1}}$), of density $\rho_s$ over a time window of length $\Delta t$. The height of snow that falls over $\Delta t$ is:

$$H_s = \frac{\mathrm{SNF}\,\Delta t}{\rho_s} \tag{B1}$$

The thickness of ice equivalent (of density $\rho_i$) is:

$$H_i = H_s \frac{\rho_s}{\rho_i} \tag{B2}$$

The equivalent thickness of air is hence $H_s - H_i$. This air thickness is not entirely available for depletion by surface melt or rainfall because a part of it remains trapped in the snow/firn/ice. Next we distinguish air depletion by rainfall and by surface melt. All the calculations below implicitly assume a positive surface mass balance with no sublimation, blowing snow or runoff. In case of negative surface mass balance, the firn air is obviously quickly depleted.

### B1 Air depletion by rainfall

We first consider a rainfall rate RF (in $\mathrm{kg\,m^{-2}\,s^{-1}}$), of density $\rho_w$ over a time window of length $\Delta t$. The thickness of rain water that falls over $\Delta t$ is:

$$H_w = \frac{\mathrm{RF}\,\Delta t}{\rho_w} \tag{B3}$$

To keep expressions simple, we assume that rainfall and snowfall are at the freezing temperature, but accounting for their temperature has a negligible effect on the estimates below (not shown).

The snow and liquid water column reaches the close-off density $\rho_{co}$ when:

$$H_w \rho_w + H_i \rho_i = H_s \rho_{co} \tag{B4}$$

Hence, rainfall saturates the snow layer, and therefore makes water available for ponding or runoff when:

$$\frac{\text{RF}}{\text{SNF}} > \frac{\rho_{co}}{\rho_s} - 1 \tag{B5}$$

For fresh snow and close-off densities of 300 and 830 $\text{kg m}^{-3}$, respectively, we get a threshold ratio of 1.77. This ratio falls to 0.66 for a snow density of 500 $\text{kg m}^{-3}$.

## B2   Air depletion by surface melt

We consider a surface melt rate MLT (in $\text{kg m}^{-2}\,\text{s}^{-1}$). The difference with the case of rainfall is that the liquid water is taken from the snowfall. Hence:

$$H_s = \frac{(\text{SNF} - \text{MLT})\,\Delta t}{\rho_s} \tag{B6}$$

and

$$H_w = \frac{\text{MLT}\,\Delta t}{\rho_w} \tag{B7}$$

In this case, Eq. (B4) gives the following condition to saturate the snow layer:

$$\frac{\text{MLT}}{\text{SNF}} > 1 - \frac{\rho_s}{\rho_{co}} \tag{B8}$$

For fresh snow and close-off densities of 300 and 830 $\text{kg m}^{-3}$, respectively, we get a threshold ratio of 0.64. This ratio falls to 0.40 for a snow density of 500 $\text{kg m}^{-3}$.

Pfeffer et al. (1991) showed that this 0.64 value was increased to $\sim$0.70 when accounting for the heat required to bring a
snow layer from $-15°$C to the freezing point. While it is clear that water cannot remain liquid in snow below the freezing point, the part of melt or rainfall that refreezes actually contributes to firn air depletion and should probably not be considered as an extra term in our calculations.

## B3   Air depletion by surface melt and rainfall

Considering the effects of melt and rain together gives the following condition to saturate the snow layer:

$$\frac{\text{MLT} + \text{RF}}{\text{SNF} - \text{MLT}} > \frac{\rho_{co}}{\rho_s} - 1 \tag{B9}$$

Importantly, the condition given by Eq. (B9) is only valid for SNF>MLT. If SNF<MLT, then it is obvious that liquid water is available for ponding or runoff. Under present and 2100-RCP8.5 conditions in the Amundsen sector, Eq. (B8) is a very good approximation of Eq. (B9) because melt rates are significantly greater than rainfall rates (see Tab. 2).

*Code availability.* The MAR code used in this study is tagged as v3.9.3 on https://gitlab.com/Mar-Group/MARv3.7. See http://www.mar.cnrs.fr for more information about downloading MAR.

*Data availability.* The present-day MAR simulation is available on http://doi.org/10.5281/zenodo.2815907. The future MAR simulation based on the CMIP5 multi-model mean is available on http://doi.org/10.5281/zenodo.4310797.

5 *Author contributions.* M. D.-M. and N. J. initiated the study, made the plots, and wrote the first draft. M. D.-M. and M. C. ran the simulations. M. D.-M., C. K., Cé. A., Ch. A. and H. G. developed the model configuration. C. K. and N. J. built the surface and lateral conditions for all the future experiments. G. K. proposed the perfect-model test. All authors took part to the result discussions and to the manuscript preparation.

*Competing interests.* The authors declare that no competing interests are present.

*Acknowledgements.* The present work was funded by the French National Research Agency (ANR) through the TROIS-AS project (ANR-
10 15-CE01-0005-01). This publication was supported by PROTECT, a project which has received funding from the European Union's Horizon 2020 research and innovation programme under grant agreement No 869304; this is PROTECT contribution number XX. All the computations presented in this paper were performed using the GRICAD infrastructure (https://gricad.univ-grenoble-alpes.fr), which is supported by Grenoble research communities.

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
