# Peer review of "Future surface mass balance and surface melt in the Amundsen sector of the West Antarctic Ice Sheet"

_The Cryosphere, 2020_

## Referee Comment (RC1) · Anonymous Referee #1 · 14 Jul 2020

Review of "Future ice-sheet surface mass balance and melting in the Amundsen region, West Antarctica", submitted for publication in The Cryosphere by M. Donat-Magnin et al.

14 July 2020

SUMMARY ===

The manuscript by Donat-Magnin describes a model experiment in which the future surface mass balance (SMB) and surface melt in the Amundsen Sea Sector of the Antarctic Ice Sheet is investigated. This is done so by using anomalies of a CMIP5 multi-model mean anomaly added to a present-day forcing to the regional climate

model MAR.

This paper is interesting both for its results and for its methodology. The Cryosphere would be a logical venue for publication, and the subject is relevant for the journal. I am enthusiastic about the methodology (including the discussion of its shortcomings) and by the results.

I would recommend publication of this paper after care has been taken to improve the manuscript on the following points.

GENERAL ===

Throughout the manuscript, the terms melt and runoff should be used with more care. It starts at P2 L18: This entire section makes sense but the use of "melt" and "runoff" needs to be more careful here. Runoff is produced only when water runs off into the ocean, and is lost from the ice sheet. In situations with ponding or hydrofracturing, leading to ice-shelf collapse, there is no runoff according to the definition, but only surface melt. If there were no ice-shelf collapse, the surface meltwater would refreeze. While runoff is currently probably about 1000 x smaller than SMB, surface melt is about 5% of SMB. It is not runoff but surface melt that triggers hydrofracturing. All over the manuscript, meltwater ponding and hydrofracturing is described as runoff, but it should be considered surface melt and not runoff, unless the water is really lost from the ice-sheet system. There are many instances with this confusion, like P10 L16 or P21 L14 and further.

SPECIFIC ===

Title: I would suggest to replace "Amundsen region" by "Amundsen sector"; remove or move the words "ice-sheet", and replace "melting" by "surface melt". So:

Future surface mass balance and surface melt in the Amundsen sector of the West Antarctic Ice Sheet.

Page 1 line 10: along -> during (or: in) P1 L11: melting -> melt P2 L2: hypothetically
stable climate -> hypothetical, stable climate P2 L8: ice cores -> firn cores P2 L15: a -> an (https://blog.apastyle.org/apastyle/2012/04/using-a-or-an-with-acronyms-and-abbreviations.html)

P2 L31: surface melting -> surface melt (everywhere in the manuscript) P2 L31: the exponential relation between temperature and surface melt is valid

P3 L13: perhaps replace the reference to Lenaerts et al. (2018) to Van Wessem et al., Modelling the climate and surface mass balance of polar ice sheets using RACMO2, part 2: Antarctica (1979–2016). The Cryosphere, 1–35 (2018). The latter is more of a benchmark publication for RACMO2.

P3 L14: slightly expand the text here to discriminate between forcing with reanalyses (Donat-Magnin, 2020, Fettweis 2013, Datta 2019) and GCMs (Trusel 2015, this paper).

P4 L16: are these sensitivities based on climatological means or instantaneous values for temperature and wind speed?

P5 L8: move this listing of models to a table or appendix.

P7 L4: remove "potential" (doubles with possible)

P10 L16: referring to the above comment, lateral transport of meltwater (into ponds for example) is not runoff in an SMB definition. If it refreezes or remains in the ice sheet it is not runoff.

P10 L22: actually it would be very interesting to show the T-M relation for your model, along with the expression from Trusel et al. It gives insight to the sensitivity of your model melt to temperature compared to previous work. Please include a figure and brief discussion here.

P10 L29: remove "also"

---

## Referee Comment (RC2) · Jan Lenaerts (Referee) · 11 Aug 2020

**Review of "Future ice-sheet surface mass balance and melting in the Amundsen region, West Antarctica" by Marion Donat-Magnin et al.**

*Jan Lenaerts and Michelle Maclennan (University of Colorado Boulder)*

This paper discusses the future SMB and components in the Amundsen Embayment area and surroundings using a regional climate model forced by bias-corrected CMIP5 forcing. The paper contains interesting and relevant results for the polar climate, firn, and ice sheet modeling communities and the topic fits very well for The Cryosphere. We have identified three major issues with the paper, as well as several minor topics for the authors to consider in a revision.

**Major issues**

1. Rainfall is not included in the analysis (Page 1, Line 13; Section 4.2), while this will be a significant component of the SMB and water budget in the ASE region in the future. Rain is highly non-linearly dependent on temperature (above the melting point), and warmer summers (and shoulder seasons) will imply more rainfall. This meltwater will need to be added to the surface meltwater to identify the liquid and solid water input to the surface.

2. (Page 2, Line 18, and many other references) In contrast to what the authors suggest, hydrofracture is not explained by/associated with runoff, but rather by a **lack** of runoff and by in-situ surface ponding of meltwater instead. Runoff is the mechanism by which water can be efficiently removed from the ice shelf, reducing hydrofracture potential. On flat ice shelves, runoff potential is limited, although local depressions on ice shelves can collect water from its surroundings, and some ice shelves have a pretty efficient wider surface drainage system (e.g. Bell et al., 2017 (Nature)). Similarly, if MAR suggests runoff to occur because a part of the surface meltwater does not refreeze, this will likely not occur in reality since the ice shelf slopes are too weak to support (widespread) lateral flow of water. This is an important misconception and needs to be addressed in the manuscript in several instances.

3. (Table 1) SMB over PIG and Thwaites are remarkably higher than obtained from extrapolating airborne radar results (Medley et al., 2014 (TC)). In their table 3, they obtain an SMB of ~67 Gt/yr over PIG and ~76 Gt/yr over Thwaites (numbers that are confirmed/validated by comparing to glacier discharge – see Figure 10 in Medley et al., 2014), suggesting that the MAR SMB is overestimated by 25-30%. This is an important bias that needs to be addressed, since it somewhat erodes the credibility of the future changes (at least in their absolute sense).

**Minor issues**

Page 1, Line 13:How well settled is this threshold, since you only use one firn model and one RCM?

Page 2, Line 20: Surface melt and/or rain

Page 2, Line 23: runoff and surface melt are used interchangeably, which is confusing. It is worth noting that surface runoff is the fraction of surface melt that does not refreeze or retain in the firn or at the surface.

Page 4, Line 25: described

Figure 3 (and others): Consider removing the southern Antarctic peninsula (and the interior ice sheet) from the figure since high SMB and melt patterns are not discussed or irrelevant in the paper, shifting the colorbar to view spatial differences in negative SMB and melt anomalies, and expressing the differences as relative instead of absolute numbers.

Table 2: consider not using 'runoff' as the generalized name for this, but rather use 'surface water budget' or something similar. As noted earlier, runoff is a fraction of and result of melt, not vice versa.

Figures: It would be very useful to add significance marking in all the maps, to highlight areas where future changes are (not) significant.

Figure 5 – remove arrows where changes are not significant.

Section 4.1: this section is very long and distracts the reader from the main message. Would it be an option to add this to an appendix or supplementary material?

---

## Author Comment (AC1) · 27 Oct 2020

SUMMARY ===

The manuscript by Donat-Magnin describes a model experiment in which the future surface mass balance (SMB) and surface melt in the Amundsen Sea Sector of the Antarctic Ice Sheet is investigated. This is done so by using anomalies of a CMIP5 multi-model mean anomaly added to a present-day forcing to the regional climate model MAR.

This paper is interesting both for its results and for its methodology. The Cryosphere would be a logical venue for publication, and the subject is relevant for the journal. I am enthusiastic about the methodology (including the discussion of its shortcomings) and by the results.

I would recommend publication of this paper after care has been taken to improve the manuscript on the following points.

→ We thank the reviewer for this positive review and careful reading. Our responses are shown in blue.

GENERAL ===

Throughout the manuscript, the terms melt and runoff should be used with more care. It starts at P2 L18: This entire section makes sense but the use of "melt" and "runoff" needs to be more careful here. Runoff is produced only when water runs off into the ocean, and is lost from the ice sheet. In situations with ponding or hydrofracturing, leading to ice-shelf collapse, there is no runoff according to the definition, but only surface melt. If there were no ice-shelf collapse, the surface meltwater would refreeze. While runoff is currently probably about 1000 x smaller than SMB, surface melt is about 5% of SMB. It is not runoff but surface melt that triggers hydrofracturing. All over the manuscript, meltwater ponding and hydrofracturing is described as runoff, but it should be considered surface melt and not runoff, unless the water is really lost from the ice- sheet system. There are many instances with this confusion, like P10 L16 or P21 L14 and further.

→ We apologize for the poor use of the term "runoff" in the submitted version of our manuscript and we understand why it may have sounded puzzling. We believe that this is a problem of terminology rather than a misunderstanding of the physical mechanisms. What we meant by "runoff" was actually the excess of meltwater and rainfall with respect to the saturation of the snow/firn column and refreezing, which could be referred to as "surface liquid water budget" or "net production of surface liquid water". Our MAR configuration removes this excess from the system (which is why we abusively called it runoff) because there is no representation of ponds or horizontal routing of liquid water. In the real world, the liquid water in excess can either form ponds, or flow horizontally toward crevasses or the ocean, but our modelling framework is not able to address the fate of this water. In our study, we used our "runoff" model variable to estimate the liquid water production beyond saturation of the snow/firn column and refreezing, not to estimate the actual runoff.

Following fair recommendations from the 3 referees, we have reformulated all the paragraphs and figures mentioning runoff.

Regarding the use of melt rate instead of runoff, we make the point in this paper that surface melt rate cannot be the relevant variable for hydrofracturing, because most melt water is retained in the annual snow layer (without ever saturating it, or, in other words, without depleting a substantial fraction of its air). Hence, in most cases, meltwater is not able to accumulate in ponds or to flow into crevasses (potentially inducing hydrofracturing) or the ocean (potentially inducing ice-shelf bending). So what we suggest in this paper is that there is only a potential for hydrofracturing ("potential" because there are also ice mechanical criteria, see, e.g., Lai et al. 2020) when the melt to snowfall ratio is high enough. Following comments from the other referees, we have also included additional comments on the role of rainfall.

We have prepared a revised version of the manuscript in which (i) we refer to the liquid water in excess as "net production of surface liquid water" instead of "runoff", and (ii) we better explain the connection with potential hydrofracturing.

SPECIFIC ===

Title: I would suggest to replace "Amundsen region" by "Amundsen sector"; remove or move the words "ice-sheet", and replace "melting" by "surface melt". So: Future surface mass balance and surface melt in the Amundsen sector of the West Antarctic Ice Sheet.
→ We have replaced with the suggested title.

Page 1 line 10: along -> during (or: in)
P1 L11: melting -> melt
P2 L2: hypothetically stable climate -> hypothetical, stable climate
P2 L8: ice cores -> firn cores
P2 L15: a -> an
→ All of these have been corrected as suggested.

P2 L31: surface melting -> surface melt (everywhere in the manuscript)
→ This has been corrected everywhere.

P2 L31: the exponential relation between temperature and surface melt is valid.
→ Ok, we have replaced "is expected to increase" with "increases".

P3 L13: perhaps replace the reference to Lenaerts et al. (2018) to Van Wessem et al., Modelling the climate and surface mass balance of polar ice sheets using RACMO2, part 2: Antarctica (1979–2016). The Cryosphere, 1–35 (2018). The latter is more of a benchmark publication for RACMO2.
→ We have added this reference.

P3 L14: slightly expand the text here to discriminate between forcing with reanalyses (Donat-Magnin, 2020, Fettweis 2013, Datta 2019) and GCMs (Trusel 2015, this paper).

→ We have slightly reformulated and expanded the text to better distinguish reanalyses and GCM forcing.

P4 L16: are these sensitivities based on climatological means or instantaneous values for temperature and wind speed?
→ They are based on instantaneous values at the time of snow deposit. This has been added.

P5 L8: move this listing of models to a table or appendix.
→ The list of models has been moved to Appendix A.

P7 L4: remove "potential" (doubles with possible)
→ We have removed "possible".

P10 L16: referring to the above comment, lateral transport of meltwater (into ponds for example) is not runoff in an SMB definition. If it refreezes or remains in the ice sheet it is not runoff.
→ This has been rephrased (see our general response).

P10 L22: actually it would be very interesting to show the T-M relation for your model, along with the expression from Trusel et al. It gives insight to the sensitivity of your model melt to temperature compared to previous work. Please include a figure and brief discussion here.
→ Here is the temperature-melt relationship in our simulations compared to Trusel's fit (dashed). In our case, each circle represents an ice-shelf grid point in the future or present-day simulation.

[Figure]

We obtain fit parameters that are slightly different from Trusel et al. (2015). However, we want equation (2) of the paper to be valid for various conditions as is the Clausius-Clapeyron formula. We have used Trusel's fit because it was calculated over 48 ice shelves all around Antarctica and is therefore more likely to remain valid far beyond the present-day

Amundsen conditions. We then use equation (3) to calculate an alternative fit of the melt to snowfall ratio based on our own simulations, which is used to estimate the uncertainty on our climate extrapolations, but we directly fit the melt to snowfall ratio.

We have added the equation of our fit in this sentence: "Recalculating an exponential fit for melt rates in a similar way as Trusel et al. (2015) also gives a stronger sensitivity (MLT = 853 $exp$(0.55 $T$)), which can be a specificity of either the Amundsen region or our model configuration." But we have not included the figure as there are already similar figures in two papers (Trusel et al. 2015 and Kuipers Munneke et al. 2014) and a lot of figures in this paper.

P10 L29: remove "also"
→ It has been removed.

---

## Author Comment (AC2) · 27 Oct 2020

**Referee #2 (Jan Lenaerts & Michelle Maclennan)**

This paper discusses the future SMB and components in the Amundsen Embayment area and surroundings using a regional climate model forced by bias-corrected CMIP5 forcing. The paper contains interesting and relevant results for the polar climate, firn, and ice sheet modeling communities and the topic fits very well for The Cryosphere. We have identified three major issues with the paper, as well as several minor topics for the authors to consider in a revision.

→ We thank Jan and Michelle for their careful review. We have addressed their three major concerns by better discussing the role of rainfall, better explaining the link between the surface liquid water budget and potential ice shelf collapse, and by showing that our present-day SMB is only weakly biased.

**Major issues**

1. Rainfall is not included in the analysis (Page 1, Line 13; Section 4.2), while this will be a significant component of the SMB and water budget in the ASE region in the future. Rain is highly non-linearly dependent on temperature (above the melting point), and warmer summers (and shoulder seasons) will imply more rainfall. This meltwater will need to be added to the surface meltwater to identify the liquid and solid water input to the surface.

→ As shown in Tab. 1, rainfall represents less than 1% of the SMB over the grounded ice sheet, even in our future RCP8.5 climate. Rainfall is more important for the liquid water budget over ice shelves than over the grounded ice sheet (Table 2), but remains of secondary importance compared to melt, representing at most 15% of melt rates, even in our RCP8.5 future projection. This was our first reason for focusing on the melt to snowfall ratio in the Discussion.

The second reason is that the rainfall to snowfall ratio has to be larger than the melt to snowfall ratio in order to deplete firn air. This is due to the fact that both melt and rainfall fill the firn porosity space, while melt additionally removes snow. Considering the simple model of Pfeffer (1991), with fresh snow of 300 kg.m$^{-3}$, firn air depletion is complete when the melt to snowfall ratio reaches 0.64. Doing a similar calculation for rainfall instead of melt gives an equivalent threshold of 1.77. This calculation has been included as Appendix B.

For these two reasons and to keep things relatively simple, we have decided to keep the discussion based on the melt to snowfall ratio. We nonetheless refer to Appendix B for more theoretical considerations on the role of rainfall, and we have added the following paragraph to the Discussion:
"The increasing proportion of liquid precipitation in a warmer climate is neglected in the above equations although it may contribute to the production of surface liquid water. Rainfall remains significantly weaker than melt rates in our RCP8.5 projections (at most 15% of melt rates in Table 2 and its capacity to deplete snow/firn air is weaker than melt rates (see Appendix B), but accounting for increasing rainfall might slightly advance the onset of net surface liquid water production late in the 22$^{nd}$ century and in the 23$^{rd}$ century. In MAR

simulations driven by CMIP6 models of high climate sensitivity, Kittel et al. (*The Cryosphere Discussion,* 2020) (their Tab. 1) found that rainfall could become as large as snowfall over the Antarctic ice shelves by the end of the 21$^{st}$ century, but corresponding melt rates would be 7 to 8 times larger than rainfall, indicating that the net production of surface liquid water remains mostly related to melt rates in conditions warmer than in our MAR projections."

2. (Page 2, Line 18, and many other references) In contrast to what the authors suggest, hydrofracture is not explained by/associated with runoff, but rather by a lack of runoff and by in-situ surface ponding of meltwater instead. Runoff is the mechanism by which water can be efficiently removed from the ice shelf, reducing hydrofracture potential. On flat ice shelves, runoff potential is limited, although local depressions on ice shelves can collect water from its surroundings, and some ice shelves have a pretty efficient wider surface drainage system (e.g. Bell et al., 2017 (Nature)). Similarly, if MAR suggests runoff to occur because a part of the surface meltwater does not refreeze, this will likely not occur in reality since the ice shelf slopes are too weak to support (widespread) lateral flow of water. This is an important misconception and needs to be addressed in the manuscript in several instances.

→ We apologize for the poor use of the term "runoff" in the submitted version of our manuscript and we understand why it may have sounded puzzling. We believe that this is a problem of terminology rather than a misunderstanding of the physical mechanisms. What we meant by "runoff" was actually the excess of meltwater and rainfall with respect to the saturation of the snow/firn column and refreezing, which could be referred to as "surface liquid water budget" or "net production of surface liquid water". Our MAR configuration removes this excess from the system (which is why we abusively called it runoff) because there is no representation of ponds or horizontal routing of liquid water. In the real world, the liquid water in excess can either form ponds, or flow horizontally toward crevasses or the ocean, but our modelling framework is not able to address the fate of this water. In our study, we used our "runoff" model variable to estimate the liquid water production beyond saturation of the snow/firn column and refreezing, not to estimate the actual runoff. Following fair recommendations from the 3 referees, we have reformulated all the paragraphs and figures mentioning runoff.

3. (Table 1) SMB over PIG and Thwaites are remarkably higher than obtained from extrapolating airborne radar results (Medley et al., 2014 (TC)). In their table 3, they obtain an SMB of ~67 Gt/yr over PIG and ~76 Gt/yr over Thwaites (numbers that are confirmed/validated by comparing to glacier discharge – see Figure 10 in Medley et al., 2014), suggesting that the MAR SMB is overestimated by 25-30%. This is an important bias that needs to be addressed, since it somewhat erodes the credibility of the future changes (at least in their absolute sense).

→ We thank the referees for pointing to Medley et al. (2014). We actually find a very good agreement with their observational estimates. The large differences between the SMB values in the present study and in Medley et al. (2014) are mostly due to different basin areas. In our study, we have used the new definition of glacial drainage basins proposed by Mouginot et al. (2017) and Rignot et al. (2019), also used in the IMBIE2 estimates. The grounded part of PIG is 186.3×10$^3$ km$^2$ in our study vs 166.8×10$^3$ km$^2$ in B. Medley's study.

Similarly, the grounded part of Thwaites is $192.4 \times 10^3$ km$^2$ for us vs $175.9 \times 10^3$ km$^2$ for B. Medley. If we scale our SMBs to match the areas used in Medley et al. (2014), we find 71.7 Gt/yr for PIG and 73.2 Gt/yr for Thwaites. These values are within the range of uncertainty of Medley et al. (67.3±6.1 Gt/yr for PIG and 75.9±5.2 Gt/yr for Thwaites).

We also would like to mention our previous study, based on the same MAR configuration (Donat-Magnin et al. 2020), and in which we assessed the simulated SMB compared to the SMB derived from airborne radar over the period 1980–2011 (Medley et al. 2013; 2014). The simulated SMB was well captured by MAR with a mean relative overestimation of approximately 10% over the Thwaites basin and local errors smaller than 20% at individual locations (Fig. 3 of Donat-Magnin et al. 2020). The interannual variability was also well simulated by MAR with a correlation of 0.90 (Fig. 4 of Donat-Magnin et al. 2020).

To better show the robustness of our simulations, we have indicated the basin areas in Table 1. We have also added a reference to Medley et al. (2014), mentioning the agreement.

**Minor issues**

Page 1, Line 13: How well settled is this threshold, since you only use one firn model and one RCM?
→ We have added "in our simulations".

Page 2, Line 20: Surface melt and/or rain
→ This has been added.

Page 2, Line 23: runoff and surface melt are used interchangeably, which is confusing. It is worth noting that surface runoff is the fraction of surface melt that does not refreeze or retain in the firn or at the surface.
→ We have rephrased all the sentences including "runoff". We now refer to the excess of liquid water as "net production of surface liquid water".

Page 4, Line 25: described
→ This has been corrected.

Figure 3 (and others): Consider removing the southern Antarctic peninsula (and the interior ice sheet) from the figure since high SMB and melt patterns are not discussed or irrelevant in the paper, shifting the colorbar to view spatial differences in negative SMB and melt anomalies, and expressing the differences as relative instead of absolute numbers.
→ We prefer keeping the entire domain (at least the part over the ice sheet, the ocean extent being already reduced) in a consistent way across all figures, and there are significant SMB changes over the interior ice sheet, which we want to show. Our colour bars have been carefully chosen to highlight patterns in the Amundsen sector. Melt patterns in the Peninsula region are often saturated (as indicated by the triangle ending of the colour bar) and. Regarding the use of relative differences, we do not consider that this would improve our figures as some areas may increase from epsilon to a few times epsilon, but still remain very low in the future.

Table 2: consider not using 'runoff' as the generalized name for this, but rather use 'surface water budget' or something similar. As noted earlier, runoff is a fraction of and result of melt, not vice versa.

→ We have rephrased all the sentences including "runoff". We now refer to the excess of liquid water as "net production of surface liquid water".

Figures: It would be very useful to add significance marking in all the maps, to highlight areas where future changes are (not) significant.

→ We have added hatches where differences are not significant in figures 3, 4 and 6.

Figure 5 – remove arrows where changes are not significant.

→ We have done as suggested.

Section 4.1: this section is very long and distracts the reader from the main message. Would it be an option to add this to an appendix or supplementary material?

→ As Referee #1, we believe that the discussion on the projection method is an important aspect of our paper, so we would like to keep it in the main part. We nonetheless understand that it distracts the reader from the story line on melt rates, firn saturation and potential for hydrofracturing. We have therefore moved the subsection of the Discussion entitled "Extrapolation to other climate perturbations" ahead of the subsection on the "Modelling and methodological limitations".

---

## Author Response (AR2)

Dear Editor,

Thank you for your work on our manuscript. We have followed your last two recommendations:
- On page 5 line 8: expand "the relaxation zone" to "the relaxation zone of the model domain"
- In figures 9 and 12, I would strongly recommend to replace abbreviations in the legend by full ice-shelf names. There is no shortage of space here, and the full names are much nicer than the abbreviations.

Best regards

Nicolas Jourdain and co-authors